# Fluorescence resonance energy transfer in atomically precise metal nanoclusters by cocrystallization-induced spatial confinement

Hao Li [1,2,3,4,6], Tian Wang [5,6], Jiaojiao Han[1,2,3,6], Ying Xu[1,2,3], Xi Kang [1,2,3] ✉, Xiaosong Li [5] ✉ & Manzhou Zhu [1,2,3] ✉

Understanding the fluorescence resonance energy transfer (FRET) of metal nanoparticles at the atomic level has long been a challenge due to the lack of accurate systems with definite distance and orientation of molecules. Here we present the realization of achieving FRET between two atomically precise copper nanoclusters through cocrystallization-induced spatial confinement. In this study, we demonstrate the establishment of FRET in a cocrystallized $Cu_8(p\text{-}MBT)_8(PPh_3)_4@Cu_{10}(p\text{-}MBT)_{10}(PPh_3)_4$ system by exploiting the overlapping spectra between the excitation of the $Cu_{10}(p\text{-}MBT)_{10}(PPh_3)_4$ cluster and the emission of the $Cu_8(p\text{-}MBT)_8(PPh_3)_4$ cluster, combined with accurate control over the confined space between the two nanoclusters. Density functional theory is employed to provide deeper insights into the role of the distance and dipole orientations of molecules to illustrate the FRET procedure between two cluster molecules at the electronic structure level.

Förster/fluorescence resonance energy transfer, a non-radiative energy transfer process, occurs through long-range dipole–dipole interactions between a donor–acceptor pair[1–3]. The term FRET is named after Theodor Förster, who proposed an equation to quantify the electronic excitation transfer efficiency from an energy donor to an acceptor, and the use of FRET as a spectroscopic or other technique has been in practice for over several decades[4]. Efficient FRET necessitates fulfilling the following conditions: (i) overlap between the emission spectrum of the donor and the excitation (or absorption) spectrum of the acceptor; (ii) small intermolecular distance between donor and acceptor; and (iii) favorable mutual orientation of their transition dipoles[5–7]. Over the past few decades, due to their ability to unravel fluorescence interactions between donor and acceptor with nanometer resolution, FRET-based sensors or imaging agents have found widespread applications in bio-related fields[8–10]. More recently, donor–acceptor composite materials have gained significant attention for their distance-dependent optoelectronic properties, which allow easy tuning of the energy transfer efficiency of the FRET system[11–14]. While FRET has been applied in various contexts, investigations into energy transfer efficiency have largely relied on semiempirical relationships[15–17]. Although the traditional FRET usually occurs based on atomically precise molecules, the relative position of molecules, the distance of molecules, and the orientation of transition dipoles were unclear in their solution systems, which hindered the directional design and modification of FRET materials.

An in-depth understanding of the energy transfer pathway at the quantum chemistry level remains challenging due to imprecise systems. In this context, the use of atomically precise systems with

[1]Department of Chemistry and Centre for Atomic Engineering of Advanced Materials, Anhui University, 230601 Hefei, China. [2]Key Laboratory of Structure and Functional Regulation of Hybrid Materials of Ministry of Education, 230601 Hefei, China. [3]Key Laboratory of Functional Inorganic Material Chemistry of Anhui Province, Anhui University, 230601 Hefei, China. [4]School of Materials and Chemical Engineering, Anhui Jianzhu University, 230601 Hefei, China. [5]Department of Chemistry, University of Washington, Seattle, WA 98195-1653, USA. [6]These authors contributed equally: Hao Li, Tian Wang, Jiaojiao Han. ✉e-mail: kangxi_chem@ahu.edu.cn; xsli@uw.edu; zmz@ahu.edu.cn

definite distance and orientation of molecules is a prerequisite for the deeply understanding of the FRET mechanism.

In the past few decades, nanoparticles have been developed as promising building blocks to construct FRET materials[18–22]. Atomically precise metal nanoclusters, a type of peculiar nanoparticles, have served as an emerging class of modular nanomaterials due to their advantage of programable geometric/electronic structures and physical/chemical properties[23–28]. Additionally, the development of metal clusters has progressed in exploring structure–property correlations at the atomic level due to their prominent quantum size effects and discrete electronic energy levels[29–34]. It has been demonstrated previously that nanoclusters can act as effective units to achieve efficient FRET[35–38]. However, for cluster-based intermolecular FRET systems, a clear perspective on the energy transfer mechanism remains inaccessible because of the imprecise structures or interactions between participating molecules[35,39,40]. Furthermore, although photoluminescence (PL) performance was exhibited in nanoclusters[41–44], accomplishing the FRET between two discrete nanocluster systems remains challenging due to their potential instability and intercluster reaction activity[45–48]. Rationally developing an atomically precise cluster-based donor–acceptor system with FRET performance allows for an in-depth understanding of the intercluster energy transfer mechanism.

Herein, the FRET was achieved between nanoclusters at the atomic level by exploiting the cocrystallization-induced spatial confinement between two fluorescent copper clusters, $Cu_8(p\text{-MBT})_8(PPh_3)_4$ (abbreviated as $Cu_8$) and $Cu_{10}(p\text{-MBT})_{10}(PPh_3)_4$ (abbreviated as $Cu_{10}$), where $p$-MBT = 4-methylbenzenethiolate. We observed the partially overlapped spectra between the emission of $Cu_8$ and the excitation of $Cu_{10}$, demonstrating their potential for constructing a cluster-based FRET system. However, the physically blended crystals of $Cu_8$ and $Cu_{10}$ clusters were still FRET inactive due to the insufficiently small intermolecular distance (Fig. 1, route I). To address this, a spatial confinement strategy, i.e., the forced cocrystallization, was exploited between $Cu_8$ and $Cu_{10}$ clusters, leading to a cocrystallized bicomponent $Cu_8(p\text{-MBT})_8(PPh_3)_4@Cu_{10}(p\text{-MBT})_{10}(PPh_3)_4$ (abbreviated as $Cu_8@Cu_{10}$). The overlapped emission of the $Cu_8$ donor and excitation of the $Cu_{10}$ acceptor, along with their controllable intermolecular distance in the cocrystallized unit cell, endowed the $Cu_8@Cu_{10}$ cocrystal with the FRET characterization (Fig. 1, route II). Both experimental efforts and theoretical calculations were performed to illustrate the FRET mechanism by investigating the nonradiative energy transfer from the $Cu_8$ donor to the $Cu_{10}$ acceptor.

## Results

### Structure and PL performance

The $Cu_8$ and $Cu_{10}$ clusters were obtained via a one-pot synthetic procedure, and their crystal structures were determined by single-crystal X-ray diffraction. Structurally, the $Cu_8$ cluster was crystallized in a triclinic $P$–1 space group (Supplementary Fig. 1 and Supplementary Table 1), and its structure could be regarded as a chair conformation $Cu_4S_2$ hexatomic ring capped by two $Cu_2S_2P_2$ motifs (Fig. 2a, b). The Cu–Cu distances ranged from 2.76 to 2.98 Å (Supplementary Table 2). The eight connective $p$-MBT ligands were bonded on the cluster surface by following two different coordination modes ($\mu_2$-S and $\mu_3$-S; Supplementary Fig. 2a). The Cu–P and Cu–S bond distances in $Cu_8$ fell in the range of 2.23–2.24 and 2.23–2.41 Å, respectively (Supplementary Table 2). The $Cu_{10}$ cluster was crystallized in a triclinic $P$–1 space group (Supplementary Fig. 3 and Supplementary Table 1), whose structure contained a rhombic $Cu_4$ ring anchored by two $Cu_3S_5P_2$ motifs at each end (Fig. 2c, d). The 10 $p$-MBT ligands also followed two different coordination modes on the $Cu_{10}$ cluster surface ($\mu_2$-S and $\mu_3$-S; Supplementary Fig. 2b). The Cu–Cu bond lengths of $Cu_{10}$ ranged from 2.74 to 2.99 Å. The Cu–P and Cu–S bond distances in $Cu_{10}$ fell in the range of 2.23–2.24 and 2.23–2.41 Å, respectively (Supplementary Table 3). The compositions of $Cu_8$ and $Cu_{10}$ clusters were further verified by electrospray ionization mass spectrometry (Supplementary Fig. 4).

The photophysical properties of $Cu_8$ and $Cu_{10}$ clusters were then investigated. The $Cu_8$ or $Cu_{10}$ clusters were non-emissive in the solution state and displayed aggregation-induced emission (AIE) with the addition of a poor solvent, i.e., methanol (Supplementary Fig. 5)[49]. In contrast, $Cu_8$ and $Cu_{10}$ clusters in the crystal state exhibited significant PL at room temperature (Supplementary Fig. 6). Accordingly, all following optical properties of these clusters were tested in their crystal state. At room temperature, the $Cu_8$ crystal displayed a maximum emission signal at 515 nm ($\lambda_{ex}$ = 365 nm; Fig. 2e). The absolute PL quantum yield (QY) at room temperature of $Cu_8$ was identified as 4.2% (Supplementary Fig. 7), and the average emission lifetime was 1.20 μs with three-lifetime components ($\tau_1$ = 4.12 μs, $\tau_2$ = 14.2 μs, and $\tau_3$ = 0.21 μs; Supplementary Fig. 8). The $Cu_8$ cluster crystal showed enhanced PL intensity and red-shifted emission spectra in wavelength from 515 to 520 nm with decreasing temperature (Supplementary

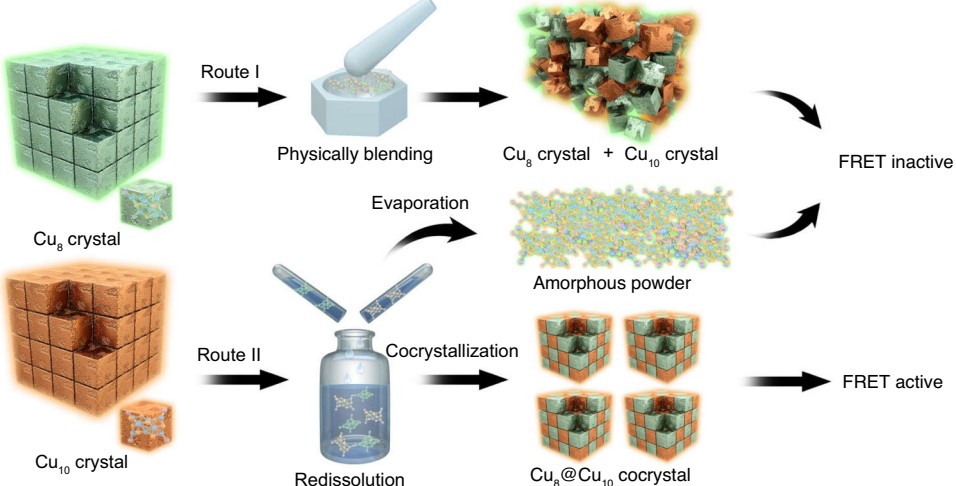

**Fig. 1 | Illustration of the rational construction of FRET-active cluster systems.** The requirements for FRET are: (i) overlapped excitation and emission and (ii) appropriate intermolecular distance. The route I represents the physically blended crystals of $Cu_8$ and $Cu_{10}$ clusters. The route II represents the forced cocrystallized $Cu_8@Cu_{10}$ cluster, which suits both requirements of FRET. Color labels: the crystals and the molecules in green represented the $Cu_8$ cluster; the crystals and the molecules in orange represented the $Cu_{10}$ cluster.

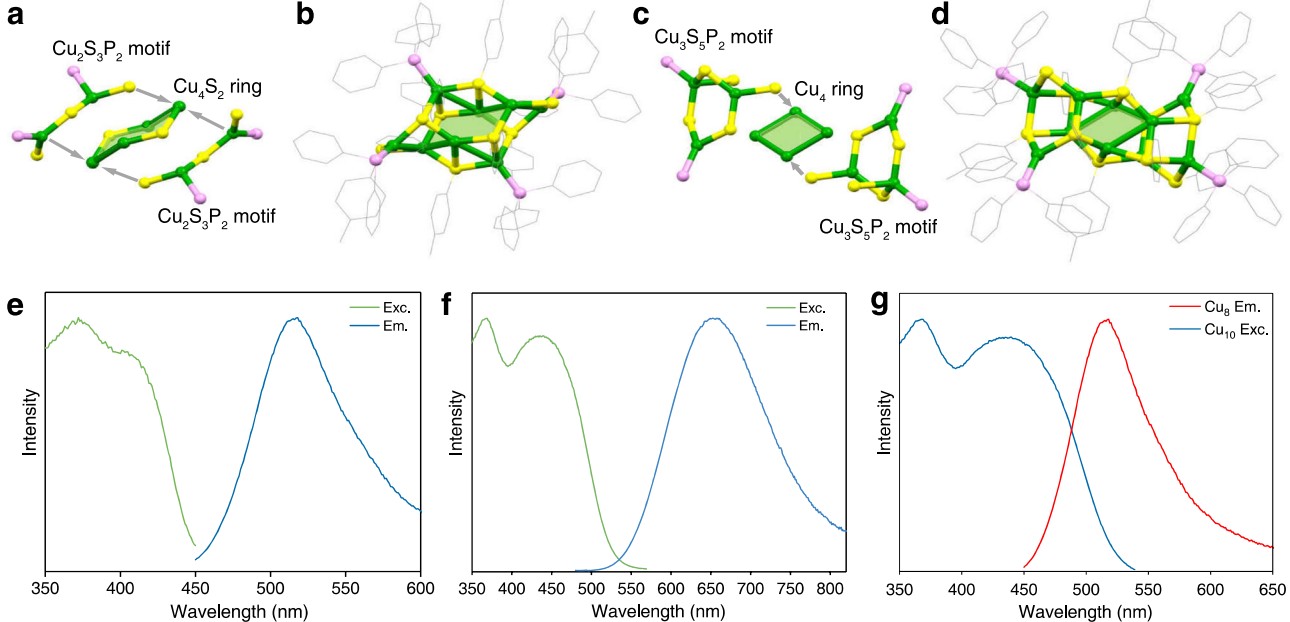

**Fig. 2 | Structure and PL performance of $Cu_8$ and $Cu_{10}$ clusters. a** Structural anatomy and **b** total structure of the $Cu_8$ cluster. **c** Structural anatomy and **d** total structure of the $Cu_{10}$ cluster. Color labels: green = Cu; yellow = S; pink = P; gray = C. All H atoms were omitted for clarity. The PL spectra of (**e**) $Cu_8$ and **f** $Cu_{10}$ clusters.

Green lines: excitation spectra; blue lines: emission spectra. **g** Spectral overlap between the excitation spectrum of $Cu_{10}$ (blue line) and the emission spectrum of $Cu_8$ (red line).

Fig. 9a, b). Although the PL changed significantly in the intensity of the $Cu_8$ nanocluster (about 53.4 times, Supplementary Fig. 9c), the PL QY did not follow the same changing trend. Indeed, the QY value was also related to the absorption of the cluster sample. The absorption spectrum of $Cu_8$ at 80 K was also measured to illustrate relative quantum yield at low temperatures. The results demonstrated that the absorption of $Cu_8$ also displayed an enhancement in intensity with the temperature decreasing (about 2.47 times, Supplementary Fig. 9d). Thus, the relative quantum yield at 80 K of $Cu_8$ nanocluster was given as 88.5%.

The crystalline state of $Cu_{10}$ clusters showed strong orange emission (QY = 41.1%, Supplementary Fig. 10) with a maximum emission wavelength at 650 nm with a microsecond emission lifetime of 5.74 μs ($\lambda_{ex}$ = 365 nm; Fig. 2f and Supplementary Fig. 11). The $Cu_{10}$ cluster exhibited enhanced PL intensity and red-shifted emission from 650 to 690 nm with the decreased temperature (Supplementary Fig. 12). Both Cu nanoclusters were stable after the temperature-dependent PL test, which displayed a similar diffraction pattern confirmed by powder X-ray diffraction (PXRD; Supplementary Fig. 13). The shift of the maximum emission wavelength was due to the alternation of the PL emission mechanism at different temperatures[49,50].

### Physical blending of crystals

We noticed that the excitation spectrum of the $Cu_{10}$ cluster overlapped with the emission spectra of the $Cu_8$ cluster at room temperature (Fig. 2g), suggesting the satisfaction of condition (i) for FRET between the $Cu_8$ (as a donor) and $Cu_{10}$ (as an acceptor) clusters. Therefore, we attempted to blend the single-component crystals of $Cu_8$ and $Cu_{10}$ clusters, and the mixture exhibited discrete PL of $Cu_8$ and $Cu_{10}$, regardless of the mole ratios between the two cluster compositions (Supplementary Fig. 14a). Due to the physical blending of $Cu_8$ and $Cu_{10}$ cluster crystals, the compositions of the mixture remained as crystals, and the molecule pair with effective FRET was not formed. In this context, the molecular distance between $Cu_8$ and $Cu_{10}$ cluster molecules was long and uncontrolled. To better control the molecular space of these two clusters, we blended the solution of $Cu_8$ and $Cu_{10}$

clusters and then made the solvent evaporation to obtain the amorphous powder solid mixture with closer intermolecular distances. The PL spectrum still displayed two-lifetime components corresponding to $Cu_8$ and $Cu_{10}$ clusters (Supplementary Fig. 14b). All these results above indicated that the PL of the $Cu_8$ cluster was not quenched in the physical mixture sample of $Cu_8$ and $Cu_{10}$ clusters. In this condition, the construction of an effective intermolecular FRET system between two clusters was unsuccessful due to the uncontrolled distance and diploe orientation between $Cu_8$ and $Cu_{10}$.

For the construction of the FRET system between $Cu_8$ and $Cu_{10}$, the key factor is to confine the distance between the two cluster molecules, i.e., to accomplish the spatial confinement between them. In previous works, although the FRET process was achieved by using sliver nanoclusters such as $Ag_{16}$ and $Ag_{29}$[35,36], the accurate dipole orientations, the favorable relative position for energy transfer, and the variation in the electronic structures of cluster molecules were hard to "see" directly in these cases. Besides, due to the various optical performances, for instance, the susceptible luminescence properties with different excitation source[51–53], the adjustable emission wavelength[54,55], and the multiple excited state[50,56], copper-based nanoclusters have been exploited as potential candidates to accomplish the FRET. Recently, increasing research has focused on the cocrystallization of heterogeneous nanoclusters[57–61]. The correlated metal/ligand compositions of $Cu_8$ and $Cu_{10}$ clusters were expected to prevent potential metal/ligand exchange reactions and form a stable coexistence system.

### Achieving FRET through cocrystallisation

The forced cocrystallization was exploited between $Cu_8$ and $Cu_{10}$ clusters, giving rise to a cocrystallized bicomponent $Cu_8(p\text{-}MBT)_8(PPh_3)_4@Cu_{10}(p\text{-}MBT)_{10}(PPh_3)_4$. Before the crystallization, the ESI-MS of the mother liquid showed a mixed composition of $Cu_8$ and $Cu_{10}$ nanoclusters (Supplementary Fig. 15). The $Cu_8@Cu_{10}$ cluster crystallizes in a triclinic $P\bar{1}$ space group with a 1:1 molecular ratio (Fig. 3a, Supplementary Fig. 16 and Supplementary Table 1). The cocrystallized $Cu_8$ and $Cu_{10}$ molecules followed a layer-by-layer

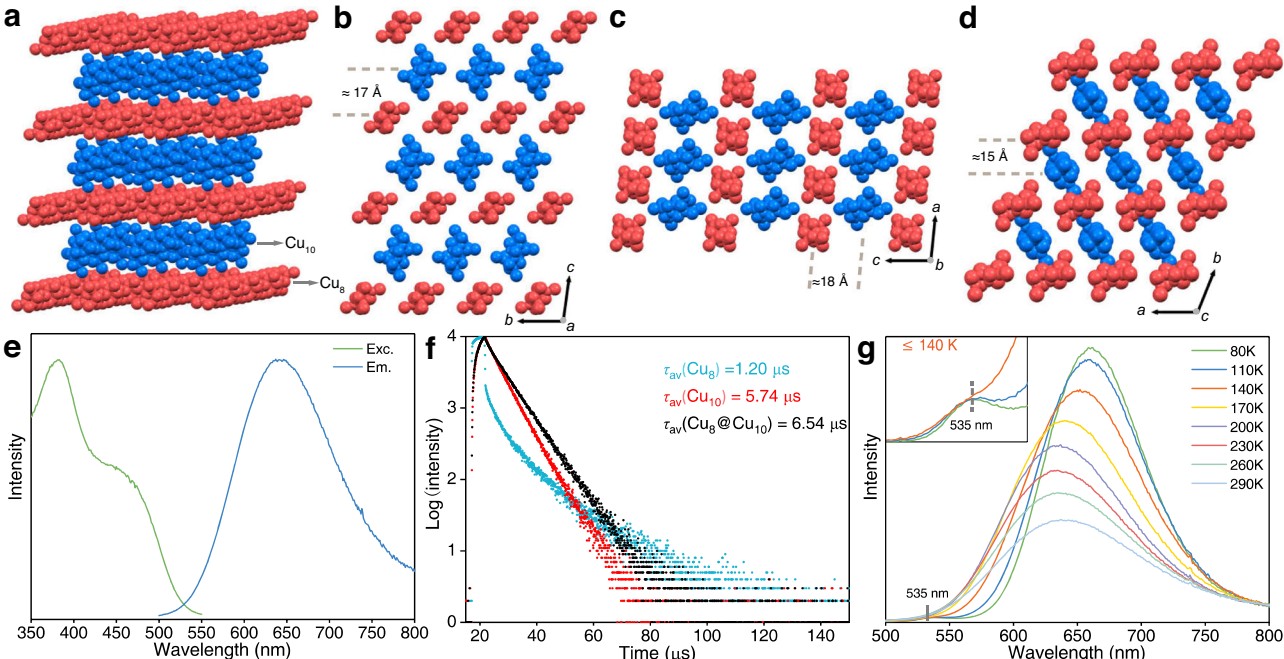

**Fig. 3 | Structure and the PL performance of Cu₈@Cu₁₀ cocrystal. a** The multi-layer structure of cocrystallized $Cu_8$ (color in red) and $Cu_{10}$ (color in blue) clusters in the crystal lattice. **b–d** Packing of nanoclusters viewed from crystallographic *a*, *b*, and *c* axes, respectively, and the molecular distance between $Cu_8$ and $Cu_{10}$ clusters. **e** The PL spectra of $Cu_8@Cu_{10}$ cocrystal at room temperature. Green lines: excitation spectra; blue lines: emission spectra. **f** The emission lifetime of $Cu_8$ (blue), $Cu_{10}$ (red), and $Cu_8@Cu_{10}$ cocrystal (black) at room temperature. **g** Temperature-dependent PL spectra of cocrystallized $Cu_8$ and $Cu_{10}$ clusters (inset: the PL spectra below 140 K).

arrangement with intermolecular distances below 2 nm (i.e., the distance of molecular center; Fig. 3c, d). The $Cu_8@Cu_{10}$ crystal displayed a strong PL at 640 nm (QY = 43.3%; Fig. 3e and Supplementary Fig. 17) with a microsecond emission lifetime of 6.54 μs with two-lifetime components ($\tau_1 = 1.72$ μs and $\tau_2 = 7.37$ μs; Fig. 3f and Supplementary Fig. 18). The single emission peak at 640 nm of the cocrystallized $Cu_8@Cu_{10}$ clusters indicated that the fluorescence of the $Cu_8$ cluster was quenched (Supplementary Fig. 19). The photophysical performance of cocrystallized $Cu_8@Cu_{10}$ was similar to that of $Cu_{10}$, indicating that the FRET process was realized. In terms of the decay time, the $Cu_8@Cu_{10}$ crystal exhibited a longer lifetime ($\tau_{av} = 6.54$ μs) than those of $Cu_8$ ($\tau_{av} = 1.20$ μs) and $Cu_{10}$ ($\tau_{av} = 5.7$ μs). The detailed photophysical data of $Cu_8$, $Cu_{10}$, and $Cu_8@Cu_{10}$ clusters at room temperature are listed in Supplementary Table 4. Temperature-dependent PL showed that the emission peak of the cocrystallized $Cu_8@Cu_{10}$ cluster at 640 nm was red-shifted to 660 nm with the temperature decreasing (Fig. 3g and Supplementary Fig. 20). The emission peak at 537 nm appeared when the temperature was below 140 K, and this peak was attributed to the $Cu_8$ cluster (inset in Fig. 3g). The emergence of the 537 nm signal at low temperatures was rational. It might be due to the following two reasons: (i) the non-radiative process in the cocrystallization system was restricted, and (ii) the radiative transition (e.g., PL) of $Cu_8$ was strengthened, which enhanced the PL QY sufficiently and the corresponding emission could be observed; indeed, the emerged 537 nm signal was similar to the emission of the monocomponent $Cu_8$ nanocluster at low temperature[62,63]. The PXRD further confirmed that the crystal structure remained unchanged after the temperature-dependent PL test (Supplementary Fig. 21). Besides, the shifted PL wavelength of the $Cu_8@Cu_{10}$ cocrystal with its single components might result from the change of the electronic structure of clusters among the intermolecular assembly, which has been observed in previous works[44,64–66]. Collectively, the FRET was realized by confining the space among cluster molecules to fix the $Cu_8$ and $Cu_{10}$ clusters in a restricted space.

## Theoretical study of FRET process

Fermi's golden rule:

$$k_{\text{FRET}} = \frac{2\pi}{\hbar}(V_{cp})^2 \text{ FCWD} \tag{1}$$

implied that the FRET rate is governed by the Franck–Condon factor weighted density of states (FCWD) realized by the spectra overlap and the electronic coupling strength ($V_{cp}$)[67]. Both key factors have been investigated by performing time-dependent density functional theory (TDDFT) calculations on $Cu_8$, $Cu_{10}$, and $Cu_8@Cu_{10}$ cocrystals, respectively.

As for the spectra overlap, the emission of $Cu_8$ (2.31 eV) centered between the absorption (2.63 eV) and emission energy (1.80 eV) of $Cu_{10}$ (Fig. 4a and Supplementary Fig. 22). The calculated absorption of $Cu_{10}$ and emission of $Cu_8$ displayed a 100 nm overlap from 450 to 550 nm, which satisfied the FRET requirement between the two clusters. The absorption and emission of $Cu_8$ corresponded to the metal-to-ligand charge transfer (MLCT) from the Cu–S backbone to $PPh_3$ ligands, and similar MLCT characterization was observed for the $Cu_{10}$ nanocluster (Supplementary Fig. 23). In the $Cu_8@Cu_{10}$ cocrystal, as shown in Supplementary Fig. 24, the MLCT-corresponded excited state ($S_{1,A}$) localized on the $Cu_{10}$ cluster molecule (HOMO-1 to LUMO), while the excited state ($S_{1,D}$) localized on the $Cu_8$ cluster molecule (HOMO to LUMO + 1). The relative energies of the frontier orbitals of the $Cu_8@Cu_{10}$ cocrystal resembled a type-II alignment (Supplementary Fig. 25)[68–70]: the HOMO of $Cu_8$ is higher than the HOMO of $Cu_{10}$ and the LUMO of $Cu_8$ is lower than the LUMO of $Cu_{10}$. The larger gap on $Cu_8$ satisfies the FRET from $Cu_8$ to $Cu_{10}$.

FRET is the energy transfer mechanism between donor and acceptor molecules. The donor ($Cu_8$), initially pumped to its electronic excited state ($S_{1,D}$), may transfer energy to excite the acceptor ($Cu_{10}$) to its excited state ($S_{1,A}$) through non-radiative coupling. The non-radiative coulombic interaction dipole-dipole between $S_{1,A}$ and $S_{1,D}$

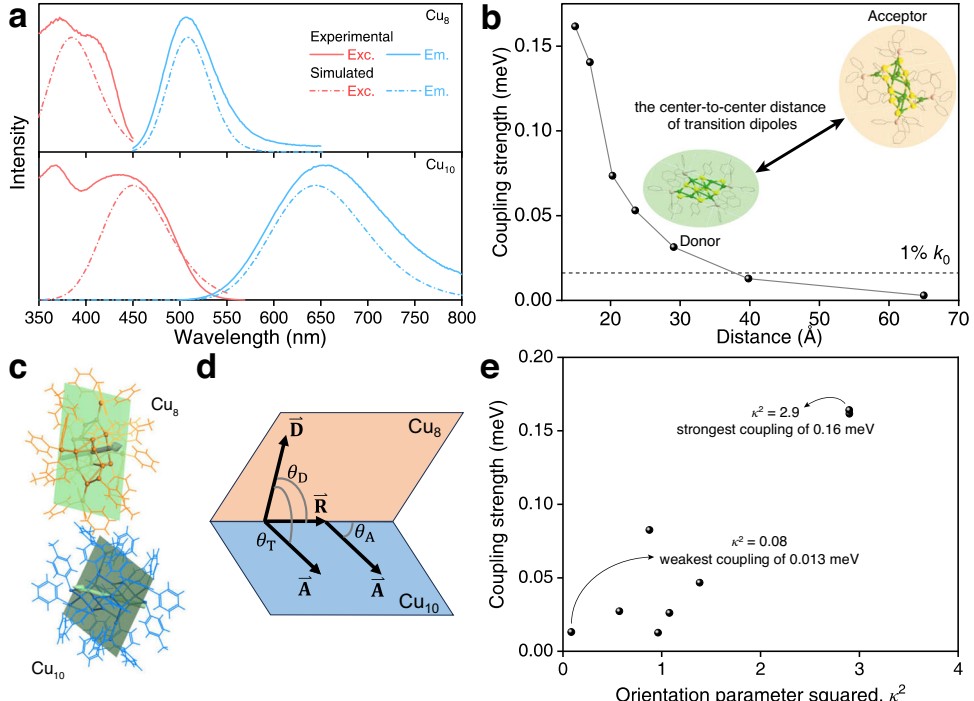

**Fig. 4 | DFT calculations for the optical spectra and coupling strength of Cu$_8$ and Cu$_{10}$ clusters. a** DFT calculated the absorption and emission spectra of Cu$_8$ or Cu$_{10}$ nanoclusters (solid lines). Dashed lines represented the experimental spectra. A uniform shift of 0.13 eV and adjusted Gaussian broadening were applied to the calculated spectra. Red lines: excitation spectra; blue lines: emission spectra. **b** The correlation between the electronic coupling strength and the center-to-center distance of transition dipoles. **c** Schematic diagram of the molecule plane for the Cu$_8$@Cu$_{10}$ cocrystal with green arrows labeling the best-fitted molecule plane. **d** Parameters for determining the orientation parameter ($\kappa$). **e** Calculated electronic coupling strength of transition dipoles with respect to the donor–acceptor orientation parameter squared ($\kappa^2$). The orientation parameter ($\kappa$) and orientation parameter squared ($\kappa^2$) are a unitless quantity.

corresponded to the FRET in the Cu$_8$@Cu$_{10}$ cocrystal. The coupling strength and estimated FRET rate are shown in Fig. 4b and Supplementary Table 5. In comparison, we investigated the direct radiative transition by evaluating the electric transition dipole moments $\langle \mathbf{i}|-\mathbf{r}|\mathbf{j}\rangle$ and its oscillator strength (a unitless quantity, the detailed value sees Supplementary Table 6). The $S_{1,A} \to S_{1,D}$ oscillator strength was only $3.00 \times 10^{-5}$. The low oscillator strength indicated a slow radiative transition rate. Therefore, the non-radiative FRET is the favored energy transfer mechanism between Cu$_8$ and Cu$_{10}$.

We also considered the possibility of Dexter energy transfer between Cu$_8$ and Cu$_{10}$ nanoclusters. Dexter energy transfer is the direct electron exchange process that requires the wavefunction overlap of HOMO (or LUMO) at the donor and acceptor, while the FRET rate is correlated with the transition dipole–dipole coupling strength (Supplementary Fig. 26). The spatial distribution of HOMO and LUMO at Cu$_8$ (donor) and Cu$_{10}$ (acceptor) is shown in Supplementary Fig. 27. The minimum distance between donor and acceptor LUMO (HOMO) is 10.7 (11.9) Å. Meanwhile, the overlap of donor and acceptor LUMO (HOMO) is negligible, indicating that the direct electron (hole) transfer is prohibited by the poor wavefunction overlap. Thus, the Dexter energy transfer is less favored in the Cu$_8$@Cu$_{10}$ cocrystal.

Collectively, the DFT calculations revealed the MLCT nature of the transition of Cu$_8$ and Cu$_{10}$ nanoclusters. Besides, the DFT calculations suggested that the FRET was induced by the energy transfer from $S_{1,D}$ localized on Cu$_8$ to $S_{1,A}$ localized on Cu$_{10}$ in the cocrystal, and a metal-to-ligand excitation on the Cu$_8$ donor and a ligand-to-metal emission on the Cu$_{10}$ acceptor was confirmed by the hole/electron spatial distribution (Supplementary Fig. 24). Therefore, the longer average PL lifetime of the cocrystallized Cu$_8$@Cu$_{10}$ than the monocomponent Cu$_8$ or Cu$_{10}$ might be attributed to it undergoing overall energy transfer processes including the excitation process of the Cu$_8$ nanocluster, the

FRET process, and the energy release process of the Cu$_{10}$ nanocluster. Based on the above results, the brief energy transfer diagram for the FRET process of the Cu$_8$@Cu$_{10}$ cocrystallized system is given in Fig. 5.

In addition, we measured $V_{cp}$ as the transition dipole–dipole interaction between $S_{1,A}$ and $S_{1,D}$ of the Cu$_8$@Cu$_{10}$ cocrystal. As shown in Fig. 4b and Supplementary Table 5, $V_{cp}$ exhibited an obvious distance dependency. The center-to-center distance of the transition dipoles (Supplementary Fig. 28) of 1.49 nm in the cocrystal corresponded to a $V_{cp}$ of 0.16 meV. The initial FRET rate $k_0$ decayed to 1% when the distance was increased to 4 nm as $k_{FRET}$ was proportional to $V_{cp}^2$, which also accounted for the decreased FRET characterization in the non-crystalline phase.

Next, we tried to determine the Förster radius ($R_0$). $R_0$ satisfied the following equation:

$$R_0 = 9.78 \times 10^3 (k^2 Q_D n^{-4} J_\lambda)^{\frac{1}{6}} \tag{2}$$

where $k^2$ is the directional relationship of transition dipoles, $Q_D$ is the quantum yield of the donor chromophore, $n$ is the refractive index of the medium, and $J_\lambda$ is the spectral overlap of the donor and acceptor. In fact, the optical spectra of the two Cu nanoclusters were different between solution and crystalline phases, which might be attributed to the variation of their electronic structures in different states. These results could be inferred from the PL and UV–vis absorption spectra. As shown in Supplementary Fig. 29, the UV–vis absorption spectra of Cu$_8$ and Cu$_{10}$ nanoclusters in CH$_2$Cl$_2$ solution display no obvious absorption band (Supplementary Fig. 29a), while strong absorptions in the crystal state were detected (Supplementary Fig. 29b). In this context, we could not deduce the parameter of the refractive index of these cluster crystals from the solution state, and thus the Förster radius was incalculable. Furthermore, we calculated the FRET rate ($k_{FRET}$) using the DFT-calculated coupling strength by exploiting

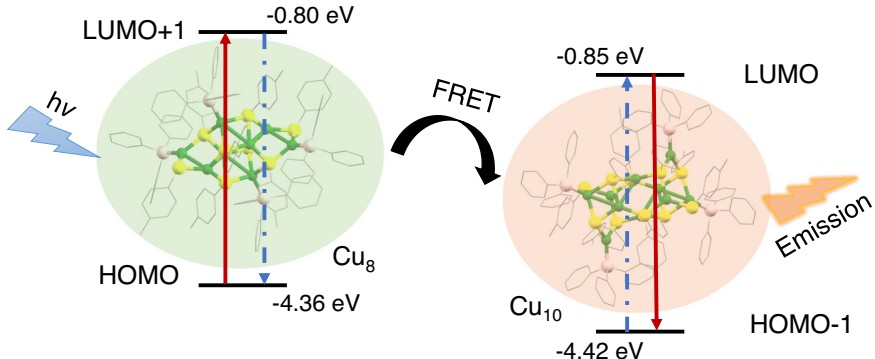

**Fig. 5 | Energy diagram for the FRET process of the Cu₈@Cu₁₀ cocrystallized system.** Color labels: green = Cu; yellow = S; pink = P; gray = C. All H atoms were omitted for clarity.

Fermi's golden rule. The FRET rate in Supplementary Table 5 is estimated by Fermi's golden rule. In addition, the FRET parameters are further estimated given that:

$$k_{\mathrm{FRET}} = k_{\mathrm{D}} \left( \frac{R_0}{r} \right)^6 \tag{3}$$

where $k_{\mathrm{D}}$ is the donor's fluorescence decay rate in the absence of the acceptor and $R_0$ is the Förster radius. Then we can further relate $V_{\mathrm{cp}}$ and $r$ as

$$V_{\mathrm{cp}}{}^2 = \frac{\hbar R_0^6 k_{\mathrm{D}}}{\mathrm{FCWD} 2\pi} r^{-6} \tag{4}$$

The linear fitting of $V_{\mathrm{cp}}{}^2$ versus $r^{-6}$ is shown in Supplementary Fig. 30. PL decay study has revealed $k_{\mathrm{D}}$ to be $2.01 \times 10^6\,\mathrm{s}^{-1}$. The FCWD was estimated to be 0.304 from the overlap of the normalized experimental spectra. Thus, the Förster radius $R_0$ was estimated to be 27.9 Å. Accordingly, the FRET efficiency ($E_{\mathrm{FRET}}$) in the different molecular distances was also given in Supplementary Table 5.

The favorable dipole orientations between the donor and the acceptor have been considered as another requirement to realize the FRET. Here, to verify the influence of dipole orientations of Cu₈ cluster donors and Cu₁₀ cluster acceptors on their FRET process, we redissolved the Cu₈@Cu₁₀ crystal after slight grinding due to the poor solubility of the crystal and then dropped the solution on the quartz plate for the solvent evaporation to form an amorphous powder. In this powder, the intermolecular distance and the dipole orientation of Cu₈ and Cu₁₀ nanoclusters were uncontrolled. Besides, the two copper nanoclusters might segregate into single separate crystal phases. In this context, the sample displayed a dual-emission spectrum corresponding to the emission of Cu₈ and Cu₁₀ clusters (Supplementary Fig. 31), indicating that the FRET was inactive in this amorphous powder.

To gain a deeper understanding of the dipole orientations between Cu₁₀ and Cu₈ and their effect on FRET, we investigated the energy transfer efficiency of each Cu₈@Cu₁₀ donor-acceptor pair with different dipole orientations in the crystal lattice. The DFT calculated $V_{\mathrm{cp}}$ was employed to characterize the FRET rate between the Cu₈ donor and the Cu₁₀ acceptor with a fixed intermolecular distance but with different dipole–dipole orientations (Fig. 4c–e). The orientations were described by the orientation parameter squared ($\kappa^2$, a unitless quantity) (Fig. 4c, d, and the detailed calculation method refers to Eq. (5) in the "Methods" section). In our simulation, $\kappa^2$ ranged from 4 (dipoles are collinear) to 0 (dipoles are perpendicular). The strongest coupling (0.16 meV) was obtained for the near-collinear orientation ($\kappa^2 = 2.9$). By comparison, the near-perpendicular orientation ($\kappa^2 = 0.08$, corresponding to a $V_{\mathrm{cp}}$ of 0.013 meV) was less favored for

FRET, resulting in a 150 times slower FRET rate ($k_{\mathrm{FRET}} \propto V_{\mathrm{cp}}{}^2$). The DFT results demonstrated that the FRET rate could be significantly affected by the intermolecular orientation since the electronic coupling strength favored the collinear transition dipole-dipole orientation. As a result, it is rational that the disordered amorphous phase of the Cu₈@Cu₁₀ nanocluster exhibited a more inactive FRET characterization relative to its cocrystals.

## Discussion

In summary, we developed a spatial confinement system, i.e., the forced cocrystallized Cu₈ and Cu₁₀ clusters, for rationally realizing the FRET in atomically precise metal nanoclusters. In contrast to the FRET inactive cluster sample of the physically blended Cu₈ and Cu₁₀ in which only the overlap between the emission of the donor and the excitation of the acceptor was achieved, the cocrystallized Cu₈@Cu₁₀ sample confined the intramolecular spaces and favored the dipole orientations between cluster donor and acceptor, resulting in the realization of the FRET between cluster molecules. In addition to the experimental efforts, theoretical calculations were performed to verify the FRET between the Cu₈ donor and the Cu₁₀ acceptor in terms of the spectra overlap, the confined space, and the dipole orientation. Overall, the spatial confinement of the cocrystallized Cu₈@Cu₁₀ cluster system presented here is of significance because it provides an ideal platform to investigate the FRET mechanism in nanomaterials.

## Methods

### Reagents

All reagents are purchased from Sigma-Aldrich and used directly without further purification: cupric acetate monohydrate [(CH₃COO)₂Cu·H₂O, 99.0%, metal basis], *p*-toluenethiol (C₇H₇S, *p*-MBT, 98%), triphenylphosphine (C₁₈H₁₅P, TPP, 99%), sodium borohydride (NaBH₄, 99%), dichloromethane (CH₂Cl₂, HPLC grade), methanol (CH₃OH, HPLC grade), *n*-hexane (C₆H₁₄, HPLC grade), and acetonitrile (CH₃CN, HPLC grade).

### Synthesis of the Cu₈(*p*-MBT)₈(TPP)₄ nanocluster

Copper acetate (0.25 mmol, 50 mg) was dissolved in 5 mL of acetonitrile, and then the solution was mixed in a round bottom flask containing 15 mL of dichloromethane. The solution was stirred vigorously at 1200 rpm. After 10 min, *p*-toluenethiol (45 mg, 0.37 mmol) was added, and the solution changed from blue-green to light yellow and turbid. After 60 min, triphenylphosphine (0.38 mmol, 100 mg) was added, and the solution gradually turned a light yellow and clarified. After 40 min, 3 mL of aqueous NaBH₄ solution (0.53 mmol mL⁻¹) was added. After 12 h, the aqueous phase was removed, and the organic phase was dried by rotary evaporation. The precipitate was dissolved with dichloromethane, and the solution was centrifuged to remove the byproducts. The yellow crystals of Cu₈(*p*-MBT)₈(TPP)₄ were obtained

by the liquid diffusion of *n*-hexane into the dichloromethane solution of the nanocluster for three days.

## Synthesis of the Cu$_{10}$(*p*-MBT)$_{10}$(TPP)$_4$ nanocluster

Copper acetate (0.2 mmol, 40 mg) was dissolved in 5 mL of methanol, and then the solution was mixed in a round bottom flask containing 15 mL of dichloromethane. The solution was stirred vigorously at 1200 rpm. After 10 min, triphenylphosphine (0.19 mmol, 50 mg) was added. Then, after 30 min, *p*-toluenethiol (0.49 mmol, 60 mg) was added, and the color of the solution changed from blue-green to light yellow. After 4 h of the reaction, the organic solvent was evaporated to half by rotary evaporation, and then 5 mL of methanol was added. The mixed solution was evaporated at 4 °C, and then orange rod-shaped crystals were obtained.

## Synthesis of the Cu$_8$(*p*-MBT)$_8$(TPP)$_4$@Cu$_{10}$(*p*-MBT)$_{10}$(TPP)$_4$ cocrystallized nanocluster

The corresponding mother solutions were obtained according to the synthetic method of Cu$_8$ and Cu$_{10}$. The mother solutions of Cu$_8$ and Cu$_{10}$ clusters were mixed, and 5 mL of methanol was added to volatilize naturally for 24 h to obtain co-crystallized Cu$_8$(*p*-MBT)$_8$(TPP)$_4$@Cu$_{10}$(*p*-MBT)$_{10}$(TPP)$_4$.

## Single-crystal X-ray diffraction

The data collection for single-crystal X-ray diffraction of two Cu$_7$ clusters was carried out on a Stoe Stadivari diffractometer under nitrogen flow using graphite-monochromatized Cu Kα radiation ($\lambda = 1.54186$ Å). Using Olex2[71], the structure was solved with the ShelXT[72] structure solution program using Intrinsic Phasing and refined with the ShelXL[73] refinement package using least-squares minimization. All the non-hydrogen atoms were found directly. All the non-hydrogen atoms were refined anisotropically. All the hydrogen atoms were set in geometrically calculated positions and refined isotropically using a riding model. The diffuse electron densities resulting from the residual solvent molecules were removed from the data set using the SQUEEZE routine of PLATON and refined further using the data generated.

## Characterization

PL spectra, absolute PL quantum yield (PL QY), and emission lifetimes were measured on a HORIBA FluoroMax-4P. The absolute PL QY test was carried out by integrating the sphere at room temperature and calculated using the FluorEssence software. The PL lifetime was fitted by the DAS6 Analysis software. The PL lifetime of the Cu$_8$ crystal was calculated by a third-order exponential fitting. The PL lifetime of Cu$_{10}$ crystals was fitted by a first-order exponential fitting. The PL lifetime of Cu$_8$@Cu$_{10}$ crystals was fitted by a second-order exponential fitting. Electrospray ionization mass (ESI-MS) was performed on Waters XEVO G2-XS QTof mass spectrometer. The samples are dissolved in a mixture solution of CH$_2$Cl$_2$/CH$_3$OH (v:v = 1:1), which is directly infused into the chamber at 10 μL min$^{-1}$ with positive mode. X-ray diffraction (XRD) pattern was obtained on SmartLab 9KW with Cu Kα radiation. UV–vis absorption spectra in the solution state were collected on a Perkin-Elmer Lambda 465 spectrophotometer. UV–vis absorption spectra in the solid state were carried on a Shimadzu 3600-plus spectrophotometer with an integrating sphere.

## DFT calculations

All DFT calculations are performed using the Gaussian 16 package[74]. The PBE0 exchange correlation is adopted for all calculations[75]. The optimization for the ground state and the excited state employed hybrid basis sets: Def2-SVP for Cu, P, and S, and SBKJC-VDZ for C and H. As for the static calculations for the absorption and emission energies, the basis set is increased to Def2-SVP for all elements. The outer D and F basis are critical for describing the Cu–P bond. The excited states are

analyzed by visualizing the spatial distribution of electrons and holes using Multiwfn[76,77]. The electron/hole isosurfaces are visualized by using VMD[78]. The absorption and emission spectra are plotted by applying a Gaussian broadening to the excitation with normalized oscillator strength. The broadening (ranging from 0.125 to 0.25 eV) is adjusted to match the experimental FWHM. The uniform broadened spectra with a narrow broadening of 0.125 eV are shown in Supplementary Fig. 22 to verify the spectra overlap and avoid manually introduced spectra overlap by over-broadening. A uniform shift of 0.13 eV is applied to the *x*-axis for all calculated spectra to be compared to the experimental spectra. The transition dipole–dipole coupling strength ($V_{cp}$) is estimated by employing the theoretical method as the formulations proposed by Iozzi, Mennucci, Tomasi, and Cammi, implemented in Gaussian 16[79]. To estimate the orientation parameter squared ($\kappa^2$), the molecule plane is defined as shown in Fig. 4c. The orientation parameter ($\kappa$, a unitless quantity) is satisfied by the equation as follows[80]:

$$\kappa^2 = \left(\cos\theta_T - 3\cos\theta_D\cos\theta_A\right)^2 \qquad (5)$$

where we define $R^{\rightarrow}$ as the vector that is orthogonal to both the normal vector of the donor plane and acceptor plane. Thus, $\theta_D$ is the angle between **R** and donor transition dipole moment. $\theta_A$ is the angle between **R** and the acceptor transition dipole moment. $\theta_T$ is the angle between the donor transition dipole and the acceptor transition dipole. The above parameters are shown in Fig. 4d.

## Data availability

The data that support the findings of this study are available from the corresponding authors upon request. Crystallographic data have been deposited at the Cambridge Crystallographic Data Centre (CCDC), under deposition numbers CCDC 2174160 (Cu$_8$), 2174162 (Cu$_{10}$), and 2174161 (Cu8@Cu$_{10}$). Cartesian coordinates for the DFT calculations, as well as cif files, have been provided as a Supplementary Data file.

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

## Acknowledgements

We acknowledge the financial support of the National Natural Science Foundation of China (21631001, 21871001, and 22101001), the Ministry of Education, and the University Synergy Innovation Program of Anhui Province (GXXT-2020-053). Computational studies are supported by the US National Science Foundation (CHE-2154346 to X.L.). Computations were facilitated through the use of advanced computational, storage, and networking infrastructure provided by the shared facility supported by the University of Washington Molecular Engineering Materials Center (DMR-2308979) via the Hyak supercomputer system. H.L. acknowledges the Scientific Research Startup Foundation of Anhui Jianzhu University (No. 2023QDZ28).

## Author contributions

H. Li conceived and carried out experiments. T. Wang carried out the DFT calculations. J. Han and Y. Xu assisted in the synthesis and optical spectral measurements. X. Kang, X. Li, and M. Zhu analyzed the data and wrote the paper. All authors contributed to the writing of the manuscript.

## Competing interests

The authors declare no competing interests.
