## [Peer Review File · Nature Communications]

Fluorescence resonance energy transfer in atomically precise metal nanoclusters by cocrystallization-induced spatial confinementREVIEWER COMMENTS

Reviewer #1 (Remarks to the Author):

See attached pdf.

In this work, the researchers synthesized and examined the photophysical properties of two copper clusters, Cu₈ and Cu₁₀. They noticed - the overlap between the excitation spectra of Cu₁₀ and the emission spectra of Cu₈, indicating the potential for Förster Resonance Energy Transfer (FRET). However, when these two clusters were physically mixed, FRET was not observed. This highlights the importance of the spatial arrangement for efficient FRET, to which, they succeeded in creating Cu₈@Cu₁₀ cocrystals. These cocrystals allowed for the precise control of intermolecular distances and favorable dipole orientations, both of which were explored through experimental and theoretical investigations. The highlight of this work lies in its synthetic methods, the comprehensive systematic analyses conducted, and the precise communication. Nevertheless, there are certain experimental concerns that require further attention.

- 1) The authors say the clusters in the solution are not emissive, Can they comment on it and is it possible to use a poor solvent and induce aggregation and obtain a similar emission behavior?
- 2) The emission spectra of both nanoclusters show a shift in their maxima at lower temperatures, the reason should be included in the ms. After the TDPL experiment is the sample still stable? PXRD must be shown.
- 3) The authors need to explain the lifetime components (instead of averaging these) of Cu₈@Cu₁₀ cocrystals and compare/comment on the changes observed with those of Cu₈ and Cu₁₀ NCs.
- 4) Can the authors comment on the change in emission from 650 to 640 nm of Cu₁₀ in Cu₈@Cu₁₀ and of Cu₈ from 515 to 537?
- 5) What is the energy transfer efficiency?
- 6) The authors have just given the lifetime number, and need to comment on the singlet or triplet origin, if it is phosphorescence the oxygen experiments need to be done, also mention in the experimental section whether the experiments are done in air/vacuum.
- 7) Are the synthesis conditions optimized? As the slight change in the amount yielded 2 different NCs? At what point they realize these changes would give rise to different NCs? What is the role of NaBH₄ in Cu₈ NC synthesis? Will it still form without adding the reducing agent?
- 8) An energy diagram showing the FRET process needs to be included for ease of understanding.
- 9) Can you assign the peak at higher m/z for both clusters? And what will be the ESI-MS spectra of Cu₈@Cu₁₀ cocrystals?
- 10) Not clear what fig S8 is telling in the ms.

Minor:

- 1) If the excitation is at 355 nm, why in Fig 2 the emission spectra is starting from 450 nm? Needs to be corrected wherever applicable.
- 2) Table S4 is not found in the MS.

Reviewer #2 (Remarks to the Author):

In this work authors have reported the realization of achieving FRET between two atomically precise Cu nanoclusters (in Cu₈@Cu₁₀ system) through spatial confinement. Also they have employed nicely DFT theory and provided deeper insights to illustrate the FRET procedure between two cluster molecules at the electronic structure level.

Results are noteworthy and will be significant for the related field. The methodology has been presented quite clearly.

The article has been written very nicely. All the data have been presented clearly. The findings in this work are very interesting and may be recommended for publication with the following revisions.

(i) The QY of the prepared materials are not very good and also widely different for two clusters. How authors have utilized it in their experiment may further be clarified. A comparison with earlier reports on QY of such NCs may be included.

The calculated QY are shown to accurate upto the 2nd decimal places. How authors have achieved such accuracy must be mentioned?

(ii) Authors have claimed that they have successfully developed a spatial confinement system. The physical mechanism of achieving the spatial confinement during the synthesis may be mentioned to make the article more acceptable/attractive to the readers.

(iii) The authors have reported that the observed energy transfer is due to FRET. Further convincing arguments may be provided on behalf of this claim.

(iv) The following articles may be included in the introduction where applications of FRET and NCs are mentioned.

<https://doi.org/10.1016/j.jcis.2020.01.019>; <https://doi.org/10.1021/acsomega.2c06011>;

<https://doi.org/10.1002/agt2.11>

Reviewer #3 (Remarks to the Author):

The manuscript by Li et. al describes experiments with atomically precise Cu clusters. Essentially they have synthesized pure Cu₈ and Cu₁₀ clusters and additionally a co-crystal between them. They perform absorption and emission experiments from which they interpret that in a co-crystal, FRET process is taking place between Cu₈ and Cu₁₀ clusters. The experiments are accompanied with DFT calculations.

The basic results of the paper, pure crystals, a co-crystal and their absorption and emission properties are interesting and worth publishing. However, I am afraid that their interpretations of the spectroscopic data are not correct and the main point of the paper is misleading. Additionally, there are many errors in the manuscript.

Main comments

1.The title is misleading in purpose. Using term "spatial confinement" gives a wrong impression of the content of the paper. It hints that the authors have created a cavity or equivalent and trapped the clusters in it. However, the spatial confinement is actually a co-crystal. The authors should use this terminology in the title.

2.In the abstract it is written "The findings in this work offer a new perspective on FRET from precise structure nanocluster systems." After reading the manuscript, it is unclear what is the new perspective here. What new on FRET was found in this paper, which was not already known? I don't understand this.

3.In the introduction, it is stated: "Precise understanding of the energy transfer pathway at the quantum chemistry level remains challenging due to imprecise systems, hindering the directional

design and modification of novel FRET materials. In this context, the use of atomically precise systems is a prerequisite for the in-depth understanding of the FRET mechanism." This is not at all correct. The authors are missing the entire history of the FRET process. FRET is very well understood nowadays both from theory and experiment. It is textbook material. One can look for example the photochemistry textbook by Turro or any other similar textbooks. There are numerous excellent papers on this. I can name two for example: *Eur. Phys. J. H.*, 39, 87 – 139 (2014), which give a historical perspective on the development of FRET, and *Annu. Rev. Phys. Chem.* 54, 57 – 87 (2003). All the excellent experimental confirmations of the FRET theory have been done at atomically precise systems, namely molecules. The current paper does not add anything fundamentally new to the field, although the system itself is an interesting case of energy transfer. It should be published as a specific case of energy transfer.

4. I think that the interpretation of the quenching observed in co-crystals is wrong. First, the radiative rate constant of the emission of Cu₈ is very small (SI Table S4). It means that the rate of FRET will be low. It probably cannot compete with nonradiative relaxation, which is much faster than emission. The authors should calculate the rate of energy transfer by using Förster theory. They have all the data needed for it (I am not sure about the molar absorption coefficient but it should be obtainable from the data). The calculation will tell if the interpretation is reasonable. The second factor speaking against FRET mechanism is the appearance of Cu₈ emission at low temperatures in the co-crystal. Why would FRET mechanism get switched off at low temperature. The statement "The appearance of the 537 nm peak was rational since the non-radiative FRET process was partially cut off at low temperatures, and then the PL of Cu₈ was recovered." does not give any explanation. In fact, a more plausible explanation for the energy transfer is the exchange or "Dexter" mechanism. This should be more effective at collision distance than FRET and especially so because the radiative rate is very low. The collision induced energy transfer may be thermally activated which gives better explanation of the temperature-dependence. So, I suspect that the whole interpretation of the paper is wrong and thus the authors should rewrite the whole story.

Other comments

5. On 5th line of introduction, the authors say "favorable mutual orientation of their dipoles" It should be transition dipoles.

6. a line above Figure 1, "nonradiative energy transformation" should be energy transfer

7. In Figure 2A, in the Cu₃S₃ ring there are 4 Cu atoms and 2 S atoms in the figure.

8. In Figure 2 E-G, Looks like the Cu₈ emission spectrum in G is not the same as the emission spectrum in E. The shape is different. Why?

9. The authors report quantum yields with two-digit accuracy, for example 4.23 % It is impossible to measure quantum yield with such accuracy. A realistic accuracy should be used. In addition, there are no details of how quantum yields were determined. It is not a trivial job. A detailed description should be provided.

10. The absence of energy transfer for mixtures of Cu₈ and Cu₁₀ is commented as "That result was rational due to the long and uncontrollable molecular distance." This is quite imprecise description. Isn't it so that it probably means that the materials don't mix at the cluster level but remain segregated as pure nano- or microcrystals in the mixture.

11. What insights do the calculations give to the FRET-process? It is not clear at all.

12. What is type-II alignment, mentioned in the paper? It is not clear for a general reader.

13. When discussing the calculated coupling constants, the authors use the distance between the nearest neighbor H-H of 0.22 nm. It is not correct as FRET refers to dipole-dipole approximation. They should rather use a distance, which corresponds to center-to-center distance of transition dipole. This is of course not exactly defined but the authors should define their choice. Then, the discussion of distances and changes in them would be more meaningful within the scope of FRET.

14. The relatively large coupling constants of 0.16 eV seem quite high considering that the experimental radiative rate constant is very low, corresponding to very low transition dipole. The authors should give their calculated transition dipoles or oscillator strengths for the relevant state and compare to experimental values. Are they anywhere close? Perhaps the states in question are not same. More details are needed.

15. Figure S4 in SI is missing y-axis scale. Is it logarithmic? This is key information. Is the x-axis correct? If the lifetime is 0.47 microseconds, i.e. 470 ns, why the emission is detected up to about 100 microseconds. There is something wrong here.

16. The caption of Figure S5 of SI doesn't make sense. "Temperature-dependent spectra at room temperature?"

Altogether, the main interpretation of the paper may be wrong, the novelty is questionable, the introduction misses the historic development of FRET and current understanding and experimental evidence of the mechanism. Additionally there are many errors and inconsistencies in the paper.

We thank all reviewers for their helpful comments. The point-by-point responses are shown in blue in this letter, and revisions are in red. Besides, the revisions in the revised manuscript are highlighted.

Reviewer #1 (Remarks to the Author):

In this work, the researchers synthesized and examined the photophysical properties of two copper clusters, Cu₈ and Cu₁₀. They noticed - the overlap between the excitation spectra of Cu₁₀ and the emission spectra of Cu₈, indicating the potential for Förster Resonance Energy Transfer (FRET). However, when these two clusters were physically mixed, FRET was not observed. This highlights the importance of the spatial arrangement for efficient FRET, to which, they succeeded in creating Cu₈@Cu₁₀ cocrystals. These cocrystals allowed for the precise control of intermolecular distances and favorable dipole orientations, both of which were explored through experimental and theoretical investigations. The highlight of this work lies in its synthetic methods, the comprehensive systematic analyses conducted, and the precise communication. Nevertheless, there are certain experimental concerns that require further attention.

Response: We thank the reviewer for the comments.

1) The authors say the clusters in the solution are not emissive. Can they comment on it and is it possible to use a poor solvent and induce aggregation and obtain a similar emission behavior?

Response: We thank the reviewer for the insightful comment. The detailed AIE performance of the Cu₈ nanocluster has been investigated in a previous work (*Angew. Chem. Int. Ed.* 2022, 61, e202200180), which demonstrated the relationship between the polymorphism-assembly of Cu₈ and emissions. We also investigated the AIE behavior of the Cu₁₀ nanocluster via adding a poor solvent (i.e., methanol), and the results indicated that the PL intensity enhanced in a certain range with the increasing volume fraction of the poor solvent. The corresponding discussions have been provided in the revised Manuscript and Supplementary Information.

We revised the Manuscript and Supplementary Information into the following statements (Pages 5&16 in the revised Manuscript; Page 3 in the revised Supplementary Information):

In the Manuscript:

(Page 5) The Cu₈ or Cu₁₀ clusters were non-emissive in the solution state and displayed aggregation-induced emission (AIE) with the addition of a poor solvent, i.e., methanol (Figure S3)⁴⁹.

(49) Sun, P.-P. *et al.* Real-Time Fluorescent Monitoring of Kinetically Controlled Supramolecular Self-Assembly of Atom-Precise Cu₈ Nanocluster. *Angew. Chem. Int. Ed.* 61, e202200180 (2022).

In the Supplementary Information:

Figure S3. AIE performance of the Cu₁₀ nanocluster in a mixed solution of CH₃OH/CH₂Cl₂ with different volume ratios (V_{CH_3OH}). The detailed AIE performance of the Cu₈ nanocluster refers to Ref: *Angew. Chem. Int. Ed.* **61**, e202200180 (2022).

2) The emission spectra of both nanoclusters show a shift in their maxima at lower temperatures, the reason should be included in the ms. After the TDPL experiment is the sample still stable? PXRD must be shown.

Response: We thank the reviewer for the professional comment. The wavelength shift in emission spectra at low temperatures was due to the change of the PL mechanism; for instance, the mixed charge-transfer (CT) including triplet metal-to-ligand or metal cluster core-to-ligand charge transfer (³MLCT or ³MMLCT) of Cu₈ cluster have been mentioned in a previous work (*Angew. Chem. Int. Ed.* **2022**, **61**, e202200180), which is also common in other Cu clusters (*Coord. Chem. Rev.*, **2018**, **377**, 307-329). In this context, the shift of the emission wavelength was caused by the alteration of the charge transfer with different temperatures. After the temperature-dependent PL tests, these crystals of Cu nanoclusters remained unchanged with maintained PXRD spectra. The corresponding discussions have been provided in the revised Manuscript and Supplementary Information.

We revised the Manuscript into the following statements (Page 5, 8, and 16 in the revised Manuscript; Pages 7&10 in the revised Supplementary Information):

In the Manuscript:

(Page 5) Both Cu nanoclusters were stable after the temperature-dependent PL test, which displayed a similar diffraction pattern confirmed by powder X-ray diffraction (PXRD; Figure S11). The shift of the maximum emission wavelength was due to the alternation of the photoluminescence emission mechanism at different temperatures^{49,50}.

(Page 8) The PXRD further confirmed that the crystal structure remained unchanged after the temperature-dependent PL test (Figure S17).

(49) Sun, P.-P. *et al.* Real-Time Fluorescent Monitoring of Kinetically Controlled Supramolecular Self-Assembly of Atom-Precise Cu₈ Nanocluster. *Angew. Chem. Int. Ed.* **61**, e202200180 (2022).

(50) Li, B., Fan, H.-T., Zang, S.-Q., Li, H.-Y. & Wang, L.-Y. Metal-containing crystalline luminescent thermochromic materials. *Coord. Chem. Rev.*, **377**, 307-329 (2018).

In the Supplementary Information:

Figure S11. Comparison of the PXRD patterns of (A) Cu₈ and (B) Cu₁₀ nanoclusters before and after the temperature-dependent PL (TDPL) test.

Figure S17. Comparison of the PXRD patterns of the cocrystallization Cu₈@Cu₁₀ before and after the TDPL test.

3) The authors need to explain the lifetime components (instead of averaging these) of Cu₈@Cu₁₀ cocrystals and compare/comment on the changes observed with those of Cu₈ and Cu₁₀ NCs.

Response: We thank the reviewer for the helpful comment. All PL lifetimes were fitted using the *DAS6 Analysis* software. The PL lifetime of the Cu₈ crystal was complicated, which was calculated by a 4-order exponential fitting equation: $A + B_1 \cdot \exp(-i/T_1) + \dots + B_4 \cdot \exp(-i/T_4)$. The result showed an average lifetime τ_{av} of 0.47 μs with four compositions ($\tau_1=0.40 \mu\text{s}$, $\tau_2=3.68 \mu\text{s}$, $\tau_3=13.85 \mu\text{s}$, and $\tau_4=0.37 \mu\text{s}$). The PL lifetime of the Cu₁₀ crystal was fitted well by 1-order exponential fitting equation ($A + B \cdot \exp(-i/T)$) and displayed a single lifetime composition ($\tau_{av}=5.74 \mu\text{s}$). The PL lifetime of the Cu₈@Cu₁₀ cocrystal was fitted by a 2-order exponential fitting equation ($A + B_1 \cdot \exp(-i/T_1) + B_2 \cdot \exp(-i/T_2)$) and displayed an average lifetime τ_{av} of 6.54 μs with two compositions ($\tau_1=1.72 \mu\text{s}$ and $\tau_2=7.37 \mu\text{s}$). Compared with the Cu₁₀ cluster, the increasing PL lifetime of Cu₈@Cu₁₀ crystals might be due to the longer energy transfer pathway from Cu₈ to Cu₁₀. Besides, a new energy diagram was added to the revised Manuscript to further illustrate the FRET process. The corresponding discussions have been added to the revised Manuscript and Supplementary Information.

We revised the Manuscript and Supplementary Information into the following statements (Pages 5, 7, and 10 in the revised Manuscript; Pages 1, 5, 6 and 9 in the revised Supplementary Information):

In the Manuscript:

(Page 5) The average emission lifetime was $0.47 \mu\text{s}$ with four compositions ($\tau_1=0.40 \mu\text{s}$, $\tau_2=3.68 \mu\text{s}$, $\tau_3=13.85 \mu\text{s}$, and $\tau_4=0.37 \mu\text{s}$; Figure S6).

(Page 7) The $\text{Cu}_8@\text{Cu}_{10}$ crystal displayed a strong PL at 640 nm (QY = $\sim 43.3\%$; Figures 3E and S14) with a microsecond emission lifetime of $6.54 \mu\text{s}$ with two compositions ($\tau_1=1.72 \mu\text{s}$ and $\tau_2=7.37 \mu\text{s}$; Figures 3F and S15).

(Page 7) Compared with the Cu_{10} cluster, the increasing PL lifetime of the $\text{Cu}_8@\text{Cu}_{10}$ cocrystallized system might result from the longer energy transfer pathway from Cu_8 to Cu_{10} nanoclusters.

(Page 10)

Figure 5. Energy diagram for the FRET process of the $\text{Cu}_8@\text{Cu}_{10}$ cocrystallized system.

In the Supplementary Information:

(Page 1) The PL lifetime was fitted by the *DAS6 Analysis* software. The PL lifetime of the Cu_8 crystal was complicated and was calculated by a 4-order exponential fitting equation: $A + B_1 \cdot \exp(-i/T_1) + \dots + B_4 \cdot \exp(-i/T_4)$. The PL lifetime of Cu_{10} crystals was fitted well by a 1-order exponential fitting equation: $A + B \cdot \exp(-i/T)$. The PL lifetime of $\text{Cu}_8@\text{Cu}_{10}$ crystals was fitted by a 2-order exponential fitting equation: $A + B_1 \cdot \exp(-i/T_1) + B_2 \cdot \exp(-i/T_2)$.

Figure S6. The emission lifetime of the Cu_8 nanocluster at room temperature.

Figure S9. The emission lifetime of the Cu₁₀ nanocluster at room temperature.

Figure S15. The emission lifetime of the Cu₈@Cu₁₀ cocrystallized system at room temperature.

4) Can the authors comment on the change in emission from 650 to 640 nm of Cu₁₀ in Cu₈@Cu₁₀ and of Cu₈ from 515 to 537?

Response: We thank the reviewer for the insightful comment. Due to the strong quantum size effect of metal nanoclusters at the nanometer level, the electronic structures are different between the single molecule and the cluster's aggregate via the distinctive packing modes, which have been demonstrated in previous works. In this context, we proposed that the different emission wavelengths of these Cu nanoclusters resulted from their changed electronic structures that led to the shift of emission wavelengths in different states. The corresponding discussions have been added to the revised Manuscript.

We revised the Manuscript into the following statements (Pages 8&16 in the revised Manuscript):

(Page 8) Besides, the shifted PL wavelength of the Cu₈@Cu₁₀ cocrystal with its single components might result from the change of the electronic structure of clusters among the intermolecular assembly, which has been observed in the previous works^{44,57-59}.

(57) Wu, Z. *et al.* Auophilic Interactions in the Self-Assembly of Gold Nanoclusters into Nanoribbons with Enhanced Luminescence. *Angew. Chem. Int. Ed.* **58**, 8139-8144 (2019).

(58) Li, H. *et al.* Triple-Helical Self-Assembly of Atomically Precise Nanoclusters. *J. Am. Chem. Soc.* **50**, 23205-23213 (2022).

(59) Nag, A. *et al.* Polymorphism of $\text{Ag}_{29}(\text{BDT})_{12}(\text{TPP})_4^{3-}$ cluster: interactions of secondary ligands and their effect on solid state luminescence. *Nanoscale* **10**, 9851-9855 (2018).

5) What is the energy transfer efficiency?

Response: We thank the reviewer for the insightful comment. The traditional FRET efficiency was calculated by the equation (*Angew. Chem. Int. Ed.*, 2016, 45, 4562-4589):

$$E = \frac{1}{1 + (r/R_0)^6}$$

where R_0 is the characteristic distance (the Förster distance or Förster radius) with a 50% transfer efficiency. R_0 satisfied the following equation:

$$R_0 = 9.78 \times 10^3 [k^2 Q_D n^{-4} J_\lambda]^{1/6}$$

where k^2 is directional relationship of transition dipoles, Q_D is quantum yield of the donor chromophore, n is the refractive index of the medium, J_λ is spectral overlap of donor and acceptor.

Different from the traditional FRET process that occurred in the solution system, the intercluster FRET process of the $\text{Cu}_8@ \text{Cu}_{10}$ cocrystallized system in this work occurred in the crystal lattice. In this work, the Förster distance (R_0) between the two Cu nanoclusters and the refractive index are all unclear. Besides, in the cocrystal, the photoluminescence behavior was not only generated from the FRET process; due to the photo-response to the exciting light of the Cu_{10} nanocluster, the PL spectra of the cocrystal contained not only the FRET process but also the emission behavior of the Cu_{10} nanocluster itself. In this context, the traditional FRET energy transfer efficiency for Cu_8 and Cu_{10} nanoclusters was difficult to calculate using the traditional Förster theory. In this work, we used the electronic coupling strength (V_{cp}) by DFT calculations to illustrate the influence of the energy transfer efficiency on the intermolecular distance and dipole orientation in the FRET process of the $\text{Cu}_8@ \text{Cu}_{10}$ cocrystallized system.

6) The authors have just given the lifetime number, and need to comment on the singlet or triplet origin, if it is phosphorescence the oxygen experiments need to be done, also mention in the experimental section whether the experiments are done in air/vacuum.

Response: We thank the reviewer for the professional comment. Because all PL experiments were carried out with crystal samples, the PL performance of these crystals was insensitive to the oxygen (previous works of singlet or triplet states focused on the PL properties in solutions). The relevant results are shown in Figure R1. Besides, the microsecond emission lifetimes of these Cu clusters suggested their triplet parentage.

Figure R1. The PL spectra of Cu cluster crystals under different atmospheres.

7) Are the synthesis conditions optimized? As the slight change in the amount yielded 2 different NCs? At what point they realize these changes would give rise to different NCs? What is the role of NaBH₄ in Cu₈ NC synthesis? Will it still form without adding the reducing agent?

Response: We thank the reviewer for the professional comment. All the synthesis conditions were optimized. The synthesis method of the Cu₈ cluster was referred to in another work (*Angew. Chem. Int. Ed.* **2022**, *61*, e202200180). If the reducing agent was absent for Cu₈ synthesis, polydisperse copper clusters/complexes would be generated. The NaBH₄ in the Cu₈ cluster synthesis might contribute to achieving a “kinetics-control & thermodynamic selection” process to form the Cu clusters. Thank you!

8) An energy diagram showing the FRET process needs to be included for ease of understanding.

Response: We thank the reviewer for the valuable suggestion. As suggested, the energy diagram of the FRET process has been added to the revised Manuscript.

We revised the Manuscript into the following statements (Page 10 in the revised Manuscript):

Figure 5. Energy diagram for the FRET process of the Cu₈@Cu₁₀ cocrystallized system.

9) Can you assign the peak at higher m/z for both clusters? And what will be the ESI-MS spectra of Cu₈@Cu₁₀ cocrystals?

Response: We thank the reviewer for the helpful comment. The higher m/z for Cu₈ and Cu₁₀ nanoclusters have been provided in the revised Supplementary Information. Due to the insolubility of Cu₈@Cu₁₀ cocrystals, we used the mother liquid before crystallization to carry out the ESI-MS measurement, and the result showed a mixed component of Cu₈ and Cu₁₀. Due to the electrically neutral molecules and the different degrees of ionization in the ion source of these two Cu clusters, although the molecule ratio is 1:1 in the unit cell, the ESI-MS result exhibited a higher response of the Cu₈ cluster than Cu₁₀ (Figure S13). The corresponding discussions have been added to the revised Manuscript and Supplementary Information.

We revised the Manuscript and Supplementary Information into the following statements (Page 7 in the revised Manuscript; Page 3&8 in the revised Supplementary Information):

In the Manuscript:

(Page 7) Before the crystallization, the ESI-MS of the mother liquid showed a mixed composition of Cu₈ and Cu₁₀ nanoclusters (Figure S13).

In the Supplementary Information:

Figure S2. ESI-MS results of Cu₈ and Cu₁₀ in the positive mode (the red line represents the simulated peak, and the black line represents the experimental peak).

Figure S13. ESI-MS result of Cu₈ and Cu₁₀ in the positive mode (the mother liquor before crystallization).

10) Not clear what fig S8 is telling in the ms.

Response: We thank the reviewer for the helpful comment. The uniform broadened spectra with narrow broadening of 0.125 eV were shown to verify the spectra overlap and to avoid manually introduced spectra overlap by over-broadening. The corresponding discussions have been added to the revised Manuscript and Supplementary Information.

We revised the Manuscript and Supplementary Information into the following statements (Page 13 in the revised Manuscript; Page 10 in the revised Supplementary Information):

In the Manuscript:

(Page 13) The uniform broadened spectra with a narrow broadening of 0.125 eV were shown in Figure S18 to verify the spectra overlap and avoid manually introduced spectra overlap by over-broadening.

In the Supplementary Information:

(Page 10) **Figure S18.** Summary of the experimental spectra of Cu₈ and Cu₁₀ nanoclusters: uniform broadened spectra with a narrow broadening of 0.125 eV.

Minor:

1) If the excitation is at 355 nm, why in Fig 2 the emission spectra is starting from 450 nm? Needs to be corrected wherever applicable.

Response: We thank the reviewer for the professional comment. As shown in Figure R2, there are no obvious PL signals before 450 nm in all emission spectra of these Cu clusters. Besides, to avoid the effect of the diploid wavelength of excitation light, a 450 nm light filter was selected to get better PL spectra of the cluster samples.

Figure R2. PL spectra of Cu clusters without the light filter.

2) Table S4 is not found in the MS.

Response: We thank the reviewer for the helpful reminder. As suggested, Table S4 has been referred to in the revised Manuscript. Thank you!

Reviewer #2 (Remarks to the Author):

In this work authors have reported the realization of achieving FRET between two atomically precise Cu nanoclusters (in Cu₈@Cu₁₀ system) through spatial confinement. Also they have employed nicely DFT theory and provided deeper insights to illustrate the FRET procedure between two cluster molecules at the electronic structure level. Results are noteworthy and will be significant for the related field. The methodology has been presented quite clearly. The article has been written very nicely. All the data have been presented clearly. The findings in this work are very interesting and may be recommended for publication with the following revisions.

Response: We thank the reviewer for supporting publication of this paper.

(i) The QY of the prepared materials are not very good and also widely different for two clusters. How authors have utilized it in their experiment may further be clarified. A comparison with earlier reports on QY of such NCs may be included. The calculated QY are shown to accurate up to the 2nd decimal places. How authors have achieved such accuracy must be mentioned?

Response: We thank the reviewer for the professional comment. The PLQY test was carried out by integrating sphere and calculated using the *FluorEssence* software. For accuracy, we kept one decimal place of the QY value in the revised manuscript. As suggested, the relevant results, methods, and QY value have been updated in the revised Manuscript and Supplementary Information.

We revised the Supplementary Information into the following statements:

(Page 1) The PLQY test was carried out by integrating sphere and calculated using the *FluorEssence* software.

Figure S5. The PLQY result of the Cu₈ nanocluster.

Figure S8. The PLQY result of the Cu₁₀ nanocluster.

Figure S14. The PLQY result of the Cu₈@Cu₁₀ cocrystal system.

(ii) Authors have claimed that they have successfully developed a spatial confinement system. The physical mechanism of achieving the spatial confinement during the synthesis may be mentioned to make the article more acceptable/attractive to the readers.

Response: We thank the reviewer for the insightful suggestion. We achieved the space limitation of Cu₈ and Cu₁₀ cluster molecules in a crystalline unit cell. Thus, the “spatial confinement” was accomplished *via* cocrystallization. We re-emphasized the method in the title of the revised Manuscript. Thank you!

(iii) The authors have reported that the observed energy transfer is due to FRET. Further convincing arguments may be provided on behalf of this claim.

Response: We thank the reviewer for the professional suggestion. In this case, all the conditions, such as the overlapping optical spectra, suitable molecular distance, and favorable dipole

orientations between the Cu₈ and Cu₁₀ clusters, were satisfied for the FRET. Besides, DFT calculations were carried out to further clarify the energy transfer pathway. In the revised Manuscript, an energy diagram was added to further illustrate the FRET process.

We revised the Manuscript into the following statements (Page 10 in the revised Manuscript):

(Page 10) Based on the above results, the brief energy transfer diagram for the FRET process of the Cu₈@Cu₁₀ cocrystallized system is given in Figure 5.

Figure 5. Energy diagram for the FRET process of the Cu₈@Cu₁₀ cocrystallized system.

(iv) The following articles may be included in the introduction where applications of FRET and NCs are mentioned. <https://doi.org/10.1016/j.jcis.2020.01.019>; <https://doi.org/10.1021/acsomega.2c06011>; <https://doi.org/10.1002/agt2.11>.

Response: We thank the reviewer for the valuable suggestion. As suggested, these relevant references have been included. Thank you!

(7) Pramanik, A. *et al.* Forster resonance energy transfer assisted white light generation and luminescence tuning in a colloidal graphene quantum dot-dye system. *J. Colloid Interface Sci.*, **565**, 326-336 (2020).

(38) Xiao, Y., Wu, Z., Yao, Q. & Xie, J. Luminescent metal nanoclusters: Biosensing strategies and bioimaging applications. *Aggregate* **2**, 114-132 (2021).

(40) Packirisamy, V. & Pandurangan, P. Interaction of Atomically Precise Thiolated Copper Nanoclusters with Proteins: A Comparative Study. *ACS Omega* **7**, 42550-42559 (2022).

Reviewer #3 (Remarks to the Author):

The manuscript by Li et. al describes experiments with atomically precise Cu clusters. Essentially they have synthesized pure Cu₈ and Cu₁₀ clusters and additionally a co-crystal between them. They perform absorption and emission experiments from which they interpret that in a co-crystal, FRET process is taking place between Cu₈ and Cu₁₀ clusters. The experiments are accompanied with DFT calculations. The basic results of the paper, pure crystals, a co-crystal and their absorption and emission properties are interesting and worth publishing. However, I am afraid that their interpretations of the spectroscopic data are not correct and the main point of the paper is misleading. Additionally, there are many errors in the manuscript.

Response: We really appreciate the reviewer's efforts in reviewing our manuscript, and providing us constructive comments and suggestions to further improve the quality of our paper. Especially, my hats go off to this reviewer who has rich knowledge on photochemistry and FRET since his/her pertinent feedbacks have helped us to significantly improve this work. The following questions have been solved point-by-point, and the Manuscript was revised accordingly. Thank you!

Main comments

1. The title is misleading in purpose. Using term "spatial confinement" gives a wrong impression of the content of the paper. It hints that the authors have created a cavity or equivalent and trapped the clusters in it. However, the spatial confinement is actually a co-crystal. The authors should use this terminology in the title.

Response: We thank the reviewer for the helpful comment. We achieved the space limitation of Cu₈ and Cu₁₀ cluster molecules in a crystalline unit cell. Thus, the "spatial confinement" was accomplished via cocrystallization. We re-emphasized the method in the title of the revised Manuscript.

The current title of this work: Rationally Realizing the Fluorescence Resonance Energy Transfer in Atomically Precise Metal Nanoclusters *via* Cocrystallization-Induced Spatial Confinement

2. In the abstract it is written "The findings in this work offer a new perspective on FRET from precise structure nanocluster systems." After reading the manuscript, it is unclear what is the new perspective here. What new on FRET was found in this paper, which was not already known? I don't understand this.

Response: We thank the reviewer for the insightful comment. For the FRET, most of previous works focused on the FRET behavior in the solution system. However, for the FRET process in the crystalline system, and for the FRET of metal nanoclusters with atomic precision, we know little about it.

In this work, we show a novel viewpoint to achieve the FRET in the crystalline unit cell *via* "spatial confinement"; specifically, the FRET occurs at the molecular level within an atomically precise system *via* fixing inter-cluster distances and orientations. Besides, based on the overlapped excitation and emission spectrum for two nanoclusters, the "spatial confinement" strategy offered a promising approach for metal nanoclusters to fabricate novel optical or energy transfer materials. Thank you!

3. In the introduction, it is stated: "Precise understanding of the energy transfer pathway at the quantum chemistry level remains challenging due to imprecise systems, hindering the directional

design and modification of novel FRET materials. In this context, the use of atomically precise systems is a prerequisite for the in-depth understanding of the FRET mechanism.” This is not at all correct. The authors are missing the entire history of the FRET process. FRET is very well understood nowadays both from theory and experiment. It is textbook material. One can look for example the photochemistry textbook by Turro or any other similar textbooks. There are numerous excellent papers on this. I can name two for example: *Eur. Phys. J. H.*, 39, 87-139 (2014), which give a historical perspective on the development of FRET, and *Annu. Rev. Phys. Chem.* 54, 57-87 (2003). All the excellent experimental confirmations of the FRET theory have been done at atomically precise systems, namely molecules. The current paper does not add anything fundamentally new to the field, although the system itself is an interesting case of energy transfer. It should be published as a specific case of energy transfer.

Response: We thank the reviewer for the professional comments. We agree with the Reviewer that FRET has been very well understood both from theory and experiment. In this work, we mainly focus on the FRET in the crystalline system of metal nanoclusters. Although the traditional FRET usually occurs based on atomically precise molecules, the relative position of molecules, the distance of molecules, and the orientation of transition dipoles were unclear in the solution system. Otherwise, we wouldn't have assumed the average value of the square of the orientation factor (κ^2) to calculate the Förster radius (R_0) and the efficiency of FRET. Once the crystal formed, all the location information of the molecules became clear, although the traditional FRET efficiency was hard to figure out. In this context, the electronic coupling strength could be utilized to describe the energy transfer efficiency in the crystal lattice via DFT calculation. We re-emphasized the distinction of the FRET between the current work and previous ones in the revised Manuscript, and some relevant References have been updated. Thank you!

We revised the Manuscript into the following statements:

(Page 2) The term FRET is named after Theodor Förster, who proposed an equation to quantify the electronic excitation transfer efficiency from an energy donor to an acceptor, and the use of FRET as a spectroscopic or other technique has been in practice for over several decades.

Although the traditional FRET usually occurs based on atomically precise molecules, the relative position of molecules, the distance of molecules, and the orientation of transition dipoles were unclear in their solution systems, which hindered the directional design and modification of novel FRET materials.

(2) Scholes, G. D. Long-Range Resonance Energy Transfer in Molecular Systems. *Annu. Rev. Phys. Chem.* **54**, 57-87 (2003).

(3) Masters, B. R. Paths to Förster's resonance energy transfer (FRET) theory. *Eur. Phys. J. H*, **39**, 87-139 (2014).

(4) Förster, T. Zwischenmolekulare Energiewanderung und Fluoreszenz. *Ann. Phys.*, **437**, 55-75 (1948).

(7) Pramanik, A. *et al.* Förster resonance energy transfer assisted white light generation and luminescence tuning in a colloidal graphene quantum dot-dye system. *J. Colloid Interface Sci.*, **565**, 326-336 (2020).

(22) Bain, D., Maity, S. & Patra, A. Opportunities and challenges in energy and electron transfer of

nanocluster based hybrid materials and their sensing applications. *Phys. Chem. Chem. Phys.* **21**, 5863-5881 (2019).

4. I think that the interpretation of the quenching observed in co-crystals is wrong. First, the radiative rate constant of the emission of Cu₈ is very small (SI Table S4). It means that the rate of FRET will be low. It probably cannot compete with nonradiative relaxation, which is much faster than emission. The authors should calculate the rate of energy transfer by using Förster theory. They have all the data needed for it (I am not sure about the molar absorption coefficient but it should be obtainable from the data). The calculation will tell if the interpretation is reasonable. The second factor speaking against FRET mechanism is the appearance of Cu₈ emission at low temperatures in the co-crystal. Why would FRET mechanism get switched off at low temperature. The statement "The appearance of the 537 nm peak was rational since the non-radiative FRET process was partially cut off at low temperatures, and then the PL of Cu₈ was recovered." does not give any explanation. In fact, a more plausible explanation for the energy transfer is the exchange or "Dexter" mechanism. This should be more effective at collision distance than FRET and especially so because the radiative rate is very low. The collision induced energy transfer may be thermally activated which gives better explanation of the temperature-dependence. So, I suspect that the whole interpretation of the paper is wrong and thus the authors should rewrite the whole story.

Response: We thank the reviewer for the professional comment. Compared with several previous works of FRET in the solution system, the Förster distance (R_0) between the two Cu nanoclusters and the refractive index of the cocrystals in this work are all unclear in the crystal state. Besides, in the cocrystal, the photoluminescence behavior was not only generated from the FRET process; due to the photo-response to the exciting light of the Cu₁₀ nanocluster, the PL spectra of the cocrystal contained not only the FRET process but also the emission behavior of the Cu₁₀ nanocluster itself. In this context, the traditional FRET energy transfer efficiency for Cu₈ and Cu₁₀ nanoclusters was difficult to calculate using the traditional Förster theory.

At low temperatures, the suppression of the non-radiative energy transfer was usually observed in previous works of metal nanoclusters. For instance, the vibration of the peripheral ligands was restricted at the low temperature so that the emission of the nanocluster enhanced; the reverse intersystem crossing (RISC) process, as a non-radiative energy transfer, in the thermally activated delayed fluorescence (TADF) was restricted at the low temperatures that would lead to the intersystem crossing (ISC) process enhancement as well the alternation of the emission mechanism. In this work, during the temperature-decreasing process, the FRET process was partially restricted so that a proportion of excited energy was released through radiative transitions (e.g., the PL) of Cu₈ and Cu₁₀ clusters. The corresponding discussions have been added in the revised Manuscript. The Dexter energy transfer is also a non-radiative process that is short-range, collisional, or exchange energy transfer with electron exchange. Usually, Dexter energy transfer is a short-range energy transfer within 10 angstroms. In this work, the center-to-center distance of the transition dipole of these two Cu nanoclusters was identified as about 15-20 angstroms, much longer than that of the Dexter energy transfer. Besides, the spatial locations of Cu₈ and Cu₁₀ nanoclusters in the cocrystal were fixed, demonstrating that the intermolecular collision was impossible. In this context, the energy transfer between the two Cu nanoclusters is more inclined to FRET than Dexter energy transfer.

We revised the Manuscript into the following statements:

(Page 8) The emergence of the 537 nm signal was rational because the non-radiative FRET process was restricted at low temperatures that was similar to previously reported ones^{55,56}, and thus the excited energy was released *via* radiative transition (e.g. PL) of Cu₈ and Cu₁₀ components, respectively.

(55) Kang, X., Wang, S. & Zhu, M. Observation of a new type of aggregation-induced emission in nanoclusters. *Chem. Sci.* **9**, 3062-3068 (2018).

(56) Han, Z. *et al.* Ultrastable atomically precise chiral silver clusters with more than 95% quantum efficiency. *Sci. Adv.* **6**, eaay0107 (2020).

Other comments:

5. On 5th line of introduction, the authors say “favorable mutual orientation of their dipoles” It should be transition dipoles.

Response: We thank the reviewer for the professional comment. We have revised the relevant content in the Manuscript.

6. a line above Figure 1, “nonradiative energy transformation” should be energy transfer.

Response: We thank the reviewer for the helpful comment. As suggested, the relevant content has been revised.

7. In Figure 2A, in the Cu₃S₃ ring there are 4 Cu atoms and 2 S atoms in the figure.

Response: We thank the reviewer for the helpful comment. This mistake has been corrected in the revised Figure 2A.

8. In Figure 2 E-G, Looks like the Cu₈ emission spectrum in G is not the same as the emission spectrum in E. The shape is different. Why?

Response: We thank the reviewer for the helpful comment. We previously used the Cu₈ cluster crystals in different synthetic batches; because of the different sizes or surface smoothness of these cluster crystals in different synthetic batches, tiny differences occurred on the peak shape; while the main emission appearances were the same, including the emission wavelength and intensity. We have used the same data of the Cu₈ emission spectrum (i.e., from the same cluster crystal) in the revised Figure 2. Thank you!

We revised the Manuscript into following statements (Page 7 in the revised Manuscript):

9. The authors report quantum yields with two-digit accuracy, for example 4.23 % It is impossible to measure quantum yield with such accuracy. A realistic accuracy should be used. In addition, there are no details of how quantum yields were determined. It is not a trivial job. A detailed description should be provided.

Response: We thank the reviewer for the insightful comment. The PLQY test was carried out by integrating sphere and calculated using the *FluorEssence* software. For accuracy, we kept one decimal place of the QY value in the revised manuscript. As suggested, the relevant results, methods, and QY value have been updated in the revised Manuscript and Supplementary Information.

We revised the Supplementary Information into the following statements:

(Page 1) The PLQY test was carried out by integrating sphere and calculated using the *FluorEssence* software.

Figure S5. The PLQY result of the Cu₈ nanocluster.

Figure S8. The PLQY result of the Cu₁₀ nanocluster.

Figure S14. The PLQY result of the Cu₈@Cu₁₀ cocrystal system.

10. The absence of energy transfer for mixtures of Cu₈ and Cu₁₀ is commented as “That result was rational due to the long and uncontrollable molecular distance.” This is quite imprecise description. Isn't it so that it probably means that the materials don't mix at the cluster level but remain segregated as pure nano- or microcrystals in the mixture.

Response: We thank the reviewer for the professional comment. Due to the physical blending of the crystals of Cu₈ and Cu₁₀, the mixed compositions remained as crystals, and the molecule pair with effective FRET was not formed. As suggested, the corresponding discussions have been revised in the Manuscript.

We revised the Manuscript into the following statements:

(Page 6) Due to the physical blending of Cu₈ and Cu₁₀ cluster crystals, the compositions of the

mixture remained as crystals, and the molecule pair with effective FRET was not formed. In this context, the molecular distance between Cu₈ and Cu₁₀ cluster molecules was long and uncontrollable.

11. What insights do the calculations give to the FRET-process? It is not clear at all.

Response: We thank the reviewer for the insightful comment. First, the DFT calculations revealed the MLCT nature of the transition of Cu₈ and Cu₁₀ nanoclusters. Second, DFT calculations demonstrated that the FRET is the energy transfer from S₂ localized on Cu₈ to S₁ localized on Cu₁₀ in the cocrystal. A metal-to-ligand excitation on the Cu₈ donor and a ligand-to-metal emission on the Cu₁₀ acceptor was confirmed by the hole/electron spatial distribution (Figure S20). Third The DFT calculations also demonstrated the distance and orientation dependency of FRET by manipulating the Cu₈ and Cu₁₀ cluster structures in the cocrystal. The corresponding discussions have been added to the revised Manuscript.

We revised the Manuscript into the following statements:

(Page 10) Collectively, the DFT calculations first revealed the MLCT nature of the transition of Cu₈ and Cu₁₀ nanoclusters. Besides, the DFT calculations suggested that the FRET was induced by the energy transfer from S₂ localized on Cu₈ to S₁ localized on Cu₁₀ in the cocrystal, and a metal-to-ligand excitation on the Cu₈ donor and a ligand-to-metal emission on the Cu₁₀ acceptor was confirmed by the hole/electron spatial distribution (Figure S20).

12. What is type-II alignment, mentioned in the paper? It is not clear for a general reader.

Response: We thank the reviewer for the helpful comment. As shown in the newly added Figure S21, three types of semiconductor heterojunctions are organized by band alignment (Adv. Mater. 2017, 29, 1601694; Angew. Chem. Int. Ed. 2012, 51, 10145). In this work, the relative energies of the frontier orbitals of the Cu₈@Cu₁₀ cocrystal resembled a Type-II alignment: HOMO of Cu₈ is higher than HOMO of Cu₁₀ and LUMO of Cu₈ is lower than LUMO of Cu₁₀. The relevant discussions have been added to the revised manuscript. We also added a schematic illustration (Figure S21) to the revised Supplementary Information to further clarify this conception.

We revised the Manuscript and Supplementary Information into the following statements:

In the Manuscript:

(Page 10) The relative energies of the frontier orbitals of the Cu₈@Cu₁₀ cocrystal resembled a Type-II alignment (Figure S21)⁶¹⁻⁶³: the HOMO of Cu₈ is higher than the HOMO of Cu₁₀ and the LUMO of Cu₈ is lower than the LUMO of Cu₁₀. The larger gap on Cu₈ satisfies the FRET from Cu₈ to Cu₁₀.

In the Supplementary Information:

Figure S21. Three types of semiconductor heterojunctions organized by band alignment. CB:

conduction band; VB: valence band.

13. When discussing the calculated coupling constants, the authors use the distance between the nearest neighbor H-H of 0.22 nm. It is not correct as FRET refers to dipole-dipole approximation. They should rather use a distance, which corresponds to center-to-center distance of transition dipole. This is of course not exactly defined but the authors should define their choice. Then, the discussion of distances and changes in them would be more meaningful within the scope of FRET.

Response: We thank the reviewer for the professional comment. We use the intermolecular distance (~ 0.22 nm) in the first version to describe the distance of the electron cloud for two Cu clusters. To better describe the distance dependency of the FRET, we approve that using the dipole-dipole distance from the center of the hole to the center of the electron in the revised Manuscript.

We revised the Manuscript and Supplementary Information into the following statements (Figure 4B and Page 10 in the revised Manuscript; Figure S22 in the revised Supplementary Information):
In the Manuscript:

Figure 4B. The correlation between the electronic coupling strength and the center-to-center distance of transition dipoles.

(Page 10) The center-to-center distance of the transition dipoles (Figure S22) of ~ 1.49 nm in the cocrystal corresponded to a V_{cp} of ~ 0.16 meV.

In the Supplementary Information:

Figure S22. Schematic diagram for defining the center-to-center distance of transition dipoles of metal nanoclusters. The arrows label the transition dipoles from the center of holes to the center of electrons. The dashed line marks the center-to-center distance of two transition dipoles.

14. The relatively large coupling constants of 0.16 eV seem quite high considering that the experimental radiative rate constant is very low, corresponding to very low transition dipole. The authors should give their calculated transition dipoles or oscillator strengths for the relevant state and compare to experimental values. Are they anywhere close? Perhaps the states in question are not same. More details are needed.

Response: We thank the reviewer for the professional comment. The DFT details of the transition for FRET, including energies, the oscillator strengths, centroid of holes/electrons, coupling strength (V_{cp}) between S_1 and S_2 , and FRET rate constant (k_{FRET}) for $Cu_8@Cu_{10}$ cocrystal have been added in Table S5 in the revised Supplementary Information. The DFT calculated oscillator strength for S_1 was 0.016, consistent with the low experimental radiative rate. The coupling strength was determined as 0.16 meV, consistent with the experimental observation. The FRET corresponded to the energy transfer from S_2 localized on Cu_8 to S_1 localized on Cu_{10} .

We revised the Supplementary Information to the following statements:

Table S5. DFT calculated excitation energies (E), oscillator strength (f), centroid of holes/electrons, coupling strength (V_{cp}) between S_1 and S_2 , and FRET rate constant (k_{FRET}) for the $Cu_8@Cu_{10}$ cocrystal.

Distance (Å)	14.9	17.0	20.3	23.5	29.1	39.8	65.0
E_{S1} (eV)	3.02	3.02	3.02	3.02	3.02	3.02	3.02
f_{S1}	0.016	0.016	0.016	0.016	0.016	0.016	0.016
E_{S2} (eV)	3.13	3.14	3.15	3.15	3.15	3.15	3.15
f_{S2}	0.007	0.007	0.008	0.008	0.009	0.009	0.009

S_1 centroid of holes (8.04, -0.03, 0.03) (7.46, 6.93, 9.78) (8.66, 8.01, 7.29) (9.78, 9.05, 4.68) (11.74, 10.92, 0.48) (15.55, 14.56, -8.09) (25.16, 23.63, 29.07)

S₁ centroid of electrons	(7.74, -0.04, -0.03)	(7.40, 6.87, 9.98)	(8.61, 7.99, 7.37)	(9.75, 9.08, 4.83)	(11.67, 10.89, 0.57)	(15.55, 14.53, -7.90)	(25.17, 23.61, 29.10)
S₂ centroid of holes	(-8.68, 0.17, -0.18)	(0.09, 0.29, 25.66)	(0.03, 0.28, 25.77)	(-0.01, 0.28, 25.81)	(-0.03, 0.16, 25.84)	(-0.02, 0.01, 25.99)	(0.02, 0.04, 26.13)
S₂ centroid of electrons	(-4.93, 4.69, -1.24)	(-0.32, 5.49, 23.55)	(-0.27, 4.88, 23.82)	(-0.20, 4.05, 24.17)	(-0.12, 2.58, 24.84)	(-0.05, 0.92, 25.60)	(0.00, 0.15, 25.93)
V_{cp} (meV)	0.162	0.140	0.074	0.053	0.031	0.013	0.003
k_{FRET} (eV/s)	7.59×10 ⁷	5.73×10 ⁷	1.57×10 ⁷	8.20×10 ⁶	2.88×10 ⁶	4.78×10 ⁵	2.43×10 ⁴

15. Figure S4 in SI is missing y-axis scale. Is it logarithmic? This is key information. Is the x-axis correct? If the lifetime is 0.47 microseconds, i.e. 470 ns, why the emission is detected up to about 100 microseconds. There is something wrong here.

Response: We thank the reviewer for the professional comment. The y-axis of all the photoluminescence lifetime spectra is logarithmic, and we have updated it in the revised Manuscript and Supplementary Information. The x-axis is calculated by the origin data by the equation (chan*time calibration). All PL lifetimes were fitted using the *DAS6 Analysis* software. The PL lifetime of the Cu₈ crystal was complicated, which was calculated by a 4-order exponential fitting equation: $A + B_1 \cdot \exp(-i/T_1) + \dots + B_4 \cdot \exp(-i/T_4)$. The result showed an average lifetime τ_{av} of 0.47 μ s with four compositions ($\tau_1=0.40 \mu$ s, $\tau_2=3.68 \mu$ s, $\tau_3=13.85 \mu$ s, and $\tau_4=0.37 \mu$ s). The PL lifetime of the Cu₁₀ crystal was fitted well by 1-order exponential fitting equation ($A + B \cdot \exp(-i/T)$) and displayed a single lifetime composition ($\tau_{av}=5.74 \mu$ s). The PL lifetime of the Cu₈@Cu₁₀ cocrystal was fitted by a 2-order exponential fitting equation ($A + B_1 \cdot \exp(-i/T_1) + B_2 \cdot \exp(-i/T_2)$) and displayed an average lifetime τ_{av} of 6.54 μ s with two compositions ($\tau_1=1.72 \mu$ s and $\tau_2=7.37 \mu$ s). In this context, although the average lifetime of the Cu₈ cluster is 0.47 μ s, it contains some longer PL lifetime. We have updated the detailed results in the revised Manuscript.

We revised the Manuscript and Supplementary Information into the following statements (Pages 5&7 in the revised Manuscript, Pages 1&4 in the revised Supplementary Information):

In the Manuscript:

(Page 5) The average emission lifetime was 0.47 μ s with four compositions ($\tau_1=0.40 \mu$ s, $\tau_2=3.68 \mu$ s, $\tau_3=13.85 \mu$ s, and $\tau_4=0.37 \mu$ s; Figure S6).

(Page 7) The Cu₈@Cu₁₀ crystal displayed a strong PL at 640 nm (QY = ~43.3 %; Figures 3E and S14) with a microsecond emission lifetime of 6.54 μ s with two compositions ($\tau_1=1.72 \mu$ s and $\tau_2=7.37 \mu$ s; Figures 3F and S15).

(Page 7) Compared with the Cu₁₀ cluster, the increasing PL lifetime of the Cu₈@Cu₁₀ cocrystallized system might result from the longer energy transfer pathway from Cu₈ to Cu₁₀ nanoclusters.

In the Supplementary Information:

(Page 1) The PL lifetime was fitted by the *DAS6 Analysis* software. The PL lifetime of the Cu₈ crystal was complicated and was calculated by a 4-order exponential fitting equation: $A + B_1 \cdot \exp(-i/T_1) + \dots + B_4 \cdot \exp(-i/T_4)$. The PL lifetime of Cu₁₀ crystals was fitted well by a 1-order exponential fitting equation: $A + B \cdot \exp(-i/T)$. The PL lifetime of Cu₈@Cu₁₀ crystals was fitted by a 2-order exponential fitting equation: $A + B_1 \cdot \exp(-i/T_1) + B_2 \cdot \exp(-i/T_2)$.

16. The caption of Figure S5 of SI doesn't make sense. "Temperature-dependent spectra at room temperature?"

Response: We thank the reviewer for the helpful comment. The caption of this Figure has been revised.

Altogether, the main interpretation of the paper may be wrong, the novelty is questionable, the introduction misses the historic development of FRET and current understanding and experimental evidence of the mechanism. Additionally there are many errors and inconsistencies in the paper.

Response: We thank the reviewer for the professional and insightful comments again. Especially, my hats go off to this reviewer who has rich knowledge on photochemistry and FRET since his/her pertinent feedbacks have helped us to significantly improve this work. As suggested, some corresponding discussions have been added in the revised Manuscript. Besides, we have re-checked the whole paper, making sure there is no error in the revised Manuscript. Thank you!

Finally, we thank all reviewers for their helpful suggestions and comments.

REVIEWER COMMENTS

Reviewer #1 (Remarks to the Author):
see attached pdf file.

In this manuscript, the authors unveil the first successful realization of Förster Resonance Energy Transfer (FRET) between two atomically precise copper nanoclusters, achieved through strategic spatial confinement. The study demonstrates FRET in a crystallized Cu₈@Cu₁₀ system, exploiting spectral overlap and precise control over nanocluster proximity. Density functional theory provides electronic structure insights into the FRET procedure.

1. **Elaborate on the Significance of Co-crystallization:** The authors are encouraged to delve deeper into the rationale behind the lack of FRET in individually mixed Cu₈ and Cu₁₀ crystals. Despite the optical properties remaining consistent, the manuscript should elucidate the importance of co-crystallization in inducing FRET, providing a comprehensive understanding of the process.
2. **Conduct Photoluminescence (PL) Studies:** In addition to lifetime data, the authors are advised to perform comprehensive PL studies on the donor molecule in the presence of the acceptor molecule. This will further elucidate the FRET mechanism, offering a more detailed analysis of the energy transfer dynamics.
3. **Quantify FRET Parameters:** Manuscript should incorporate the calculation of crucial FRET parameters, including overlap integral, energy transfer efficiency, Forster radius, and the energy transfer rate.
4. **Highlight Advantages over Previous Studies:** The authors should dedicate a section to discussing the advantages of the FRET process based on copper nanoclusters compared to previously reported silver nanoclusters. This comparative analysis will underscore the uniqueness and advancements offered by the current research.
5. **Cite Relevant Copper Nanocluster Literature:** Strengthen the manuscript's foundation by citing more papers related to copper nanoclusters.

Addressing these points will further solidify the manuscript's significance, making it suitable for publication in Nature Communications.

Reviewer #2 (Remarks to the Author):

Authors have made substantive revisions based on the comments of the reviewers. The manuscript is now recommended for its acceptance for publication.

Reviewer #3 (Remarks to the Author):

The authors have answered many questions satisfactorily and the manuscript has been improved. However, there are still some unanswered questions and errors that needs to be solved, and some important information is missing.

1. There is still no explanation for the suppression of FRET at low temperatures. In new sentence "The emergence of the 537 nm signal was rational because the non-radiative FRET process was restricted at low temperatures that was similar to previously reported ones 55,56...", a reference is made to previous studies, where emission from similar systems is changed at low temperatures, and it is explained in terms of thermally activated delayed fluorescence or suppression of non-radiative pathways via freezing of vibrational degrees of freedom. For FRET, similar mechanism does not apply as it does not need any thermal excitation. An alternative explanation must be looked for. Perhaps one possibility is that at low temperature, the non-radiative relaxation rate of the Cu8 cluster decreases sufficiently to make QY for emission large enough to be observed. This is in line with Fig S7. After all, the observed emission intensity depends on all competing rates. The authors should elaborate more on this and either give a plausible explanation of the phenomenon, or admit that they don't have an explanation on it.

Additionally, there is repetition in the text: "and thus the excited energy was released via radiative transition (e.g. PL) of Cu8 and Cu10 components, respectively, and then the energy released via radiative transition (e.g. PL) of Cu8 and Cu10 clusters, respectively."

2. I would still recommend to perform a Förster theory calculation of the energy transfer rate using the available data. As the spectra don't change radically from solution to crystalline phase, the states should be reasonably similar and the calculation should give a correct order-of-magnitude estimate, thus making an important contribution to the paper. However, as there is new computational data on energy transfer rate, I would not consider this critical.

3. I have a comment on the answer of the authors to the comment 4, regarding the possibility of the Dexter mechanism. The authors state that "the center-to-center distance of the transition dipole of these two Cu nanoclusters was identified as about 15-20 angstroms, much longer than that of the Dexter energy transfer. Besides, the spatial locations of Cu8 and Cu10 nanoclusters in the cocrystal were fixed, demonstrating that the intermolecular collision was impossible. In this context, the energy transfer between the two Cu nanoclusters is more inclined to FRET than Dexter energy transfer." However, It is not true that intermolecular collisions are not possible in a crystal. Near neighbors undergo thermally activated collisions, which modulate for example the Dexter energy transfer rate. Additionally, For Dexter mechanism, the center to center distance of the transition dipoles is not a proper parameter. Dexter mechanism requires overlap of electronic orbitals. Thus, the relevant parameter is the distance compared to the Van der Waals distance (see, for example the Turro book). The authors may want to once more consider the possibility of the Dexter mechanism in this light.

4. The calculations suggest that the energy transfer occurs from the S2 state of the Cu8 cluster. In this case, isn't the non-radiative relaxation from S2 to S1, which should be very rapid, a problem? What is the rate for this relaxation? There should be information on the properties of the S2 state and a comment on this question.

5. There are inconsistencies regarding the electronic coupling strength. In the rebuttal letter, the coupling is given in eV as well as in the Fig 4E of the main article, whereas in text and Fig. 4B, it is

given in meV. These inconsistencies should be corrected. I suppose meV is correct.

6. The fitting data on Fig. S6 seems strange. First of all, there are two almost equal lifetime components (0.4 and 0.37 microseconds). The small difference indicates that three components should be sufficient. The authors should try that. Second, it is not clear how the average lifetime of 0.47 microseconds was obtained. Is it an amplitude weighted average of all the lifetimes? This should be explained. In any case, the average lifetime seems quite low. In Fig. S6, the scale of the y-axis should be included (also in S9). Then it will be possible to judge if the average lifetime reflects the time evolution correctly.

Additionally, what is the instrumental response time? Is it possible that the shortest lifetime component actually is giving the instrumental response time?

7. In Figs. S9 and S15, there seems to be a rise time component. Is that correct. It should be mentioned and its significance briefly discussed.

8. In the introduction, the statement: "Herein, the FRET was achieved for the first time between nanoclusters at the atomic level by exploiting..." is not justified. I think there are previous studies of FRET between atomically precise nanoclusters. The sentence should be modified.

9. For the Cu₈ cluster, the emission QY is stated as 4.2 %. Is that measured at room temperature? This should be given. The temperature dependence seems to be very strong especially between 200 and 230 K (Fig. S8). It would be interesting to see the T-dependence plotted. Is there an Arrhenius-type of behavior?

Although it is very hard to see from Fig. S7, it seems that the photoluminescence intensity increases by a factor much larger than 20, perhaps by 100, when going from the room temperature to 80 K and presumably it would increase further at lower temperatures. Referring to point 9, if the QY is 4.2 % at RT, the large increase of intensity would mean >100 % QY at low T. This possible discrepancy should be considered and explained.

10. In Fig. 4B, it would be interesting to see if the calculated energy transfer rate follows $1/r^6$ distance-dependence, as it should. Perhaps a fit could be tried and possibly included in the Figure.

11. The interpretation of the amorphization experiment (observation of dual emission) of the co-crystal does not sound right. It seems highly unlikely that the orientation factor could explain the suppression of energy transfer. I suppose a better explanation is that the individual clusters are again segregating into separate phases. In an amorphous crystal, there would in any case always exist favorable directions for energy transfer between some pairs of nearest neighbor clusters.

12. When describing the lifetime fitting results, instead of using terminology "two compositions" it would be better to use "two lifetime components".

We thank all reviewers for their helpful comments. The point-by-point responses are shown in blue in this letter, and revisions are in red. Besides, the revisions in the revised manuscript are highlighted.

Reviewer #1 (Remarks to the Author):

In this manuscript, the authors unveil the first successful realization of Förster Resonance Energy Transfer (FRET) between two atomically precise copper nanoclusters, achieved through strategic spatial confinement. The study demonstrates FRET in a crystallized Cu₈@Cu₁₀ system, exploiting spectral overlap and precise control over nanocluster proximity. Density functional theory provides electronic structure insights into the FRET procedure.

Response: We thank the reviewer for the professional comments.

1. Elaborate on the Significance of Co-crystallization: The authors are encouraged to delve deeper into the rationale behind the lack of FRET in individually mixed Cu₈ and Cu₁₀ crystals. Despite the optical properties remaining consistent, the manuscript should elucidate the importance of co-crystallization in inducing FRET, providing a comprehensive understanding of the process.

Response: We thank the reviewer for insightful comments. Although previous studies have accomplished the FRET with the participation of atomically precise nanoclusters, these cases were almost achieved in solutions. In this work, we realized the FRET between nanoclusters at the atomic level *via* cocrystallization-induced spatial confinement strategy; that is, the FRET was accomplished in the crystal system. We have re-emphasized the strategy in the revised Manuscript. Thank you!

We revised the Manuscript into the following statements:

(Page 3) Herein, the FRET was achieved for the first time between nanoclusters at the atomic level by exploiting the cocrystallization-induced spatial confinement between two fluorescent copper clusters.

2. Conduct Photoluminescence (PL) Studies: In addition to lifetime data, the authors are advised to perform comprehensive PL studies on the donor molecule in the presence of the acceptor molecule. This will further elucidate the FRET mechanism, offering a more detailed analysis of the energy transfer dynamics.

Response: We thank the reviewer for the professional comment. Because the FRET only occurred in the cocrystallized Cu₈@Cu₁₀ system in our work, we use DFT to further illustrate the FRET mechanism. More in-depth discussions have been added to the Manuscript. Thank you!

We revised the Manuscript and the Supplementary Information into the following statements:

(Page 10) FRET is the energy transfer mechanism between donor and acceptor molecules. The donor (Cu₈), initially pumped to its electronic excited state (S₂), may transfer energy to excite the acceptor (Cu₁₀) to its excited state (S₁) through non-radiative coupling. The non-radiative Coulombic interaction dipole-dipole between S₁ and S₂ corresponded to the FRET in the Cu₈@Cu₁₀ cocrystal. The coupling strength and estimated FRET rate are shown in Figure 4B and Table S5. In comparison, we investigated the direct radiative transition by evaluating the electric transition dipole moments $\langle i | -r | j \rangle$ and its oscillator strength (Table S6). The S₁→S₂ oscillator strength was only 3.00×10⁻⁵. The low oscillator strength indicated a slow radiative transition rate. Therefore, the

non-radiative FRET is the favored energy transfer mechanism between Cu₈ and Cu₁₀.

(Page 10) We also considered the possibility of Dexter energy transfer between Cu₈ and Cu₁₀ nanoclusters. Dexter energy transfer is the direct electron exchange process that requires the wavefunction overlap of HOMO (or LUMO) at the donor and acceptor, while the FRET rate is correlated with the transition dipole-dipole coupling strength (Figure S22). The spatial distribution of HOMO and LUMO at Cu₈ (donor) and Cu₁₀ (acceptor) is shown in Figure S23. The minimum distance between donor and acceptor LUMO (HOMO) is 10.7 (11.9) Å. Meanwhile, the overlap of donor and acceptor LUMO (HOMO) is negligible, indicating that the direct electron (hole) transfer is prohibited by the poor wavefunction overlap. Thus, the Dexter energy transfer is less favored in the Cu₈@Cu₁₀ cocrystal.

(Page 11) Therefore, the longer average PL lifetime of cocrystallized Cu₈@Cu₁₀ crystal than the monocomponent Cu₈ or Cu₁₀ nanocluster might be attributed to it undergoing energy transfer processes including the excitation process of the Cu₈ nanocluster, the FRET process, and the energy release process of the Cu₁₀ nanocluster.

In the Supplementary Information:

Figure S22. Schematic process of Förster resonance energy transfer and Dexter energy transfer.

Figure S23. Spatial distribution of HOMO and LUMO on donor (Cu_8) and acceptor (Cu_{10}).

3. **Quantify FRET Parameters:** Manuscript should incorporate the calculation of crucial FRET parameters, including overlap integral, energy transfer efficiency, Forster radius, and the energy transfer rate.

Response: We thank the reviewer for the professional comment. The key issue in solving the FRET efficiency is knowing the Förster radius (R_0). In fact, the optical spectra of the two Cu nanoclusters were inconsistent between solution and crystalline phases, which might be attributed to their different electric structures in the two states. These results could be inferred from the PL and UV-vis absorption spectra. As shown in Figure R1, the UV-vis absorption spectra of Cu_8 and Cu_{10} nanoclusters in CH_2Cl_2 solution display no obvious absorption band (Figure R1A), while strong absorptions in the crystal state were detected (Figure R1B). In this context, we couldn't deduce the parameter of the refractive index of these cluster crystals from the solution state, and thus the Förster radius was incalculable *via* the following equation:

$$R_0 = 9.78 \times 10^3 [k^2 Q_D n^{-4} J_\lambda]^{1/6}$$

where k^2 is the directional relationship of transition dipoles, Q_D is the quantum yield of the donor chromophore, n is the refractive index of the medium, and J_λ is the spectral overlap of the donor and acceptor.

Figure R1. The UV-Vis spectra of these Cu nanoclusters in (A) solution and (B) crystal states.

We calculated the FRET rate (k_{FRET}) using the DFT calculated coupling strength by exploiting the Fermi's golden rule $k_{\text{FRET}} = \frac{2\pi}{\hbar} [V_{cp}]^2 \text{FCWD}$ (Table S5). In addition, the FRET parameters are further estimated given that $k_{\text{FRET}} = k_D \left(\frac{R_0}{r}\right)^6$, where k_D is the donor's fluorescence decay rate in the absence of the acceptor and R_0 is the Förster radius. Then we can further relate V_{cp} and r as: $[V_{cp}]^2 = \frac{\hbar R_0^6 k_D}{\text{FCWD} 2\pi} r^{-6}$. The linear fitting of $[V_{cp}]^2$ versus r^{-6} is shown in Figure R2. PL decay study has revealed k_D to be 5.12×10^6 eV/s. The Franck-Condon factor weighted density of states (FCWD) is estimated to be 0.304 from the overlap of the normalized experimental spectra. Thus, the Förster radius R_0 is estimated to be 23.8 Å. Accordingly, the FRET efficiency (E) in the different molecular distance was also given via the following equation:

$$E = \frac{1}{1 + (r/R_0)^6}$$

where R_0 is the Förster radius with a 50% transfer efficiency.

Figure R2. Square of electronic coupling strength ($[V_{cp}]^2$) with respect to the -6 power of the distance (r^{-6}).

We revised the Manuscript and the Supplementary Information into the following statements:

In the Manuscript:

(Page 11) Next, we tried to determine the Förster radius (R_0). R_0 satisfied the following equation:

$$R_0 = 9.78 \times 10^3 [k^2 Q_D n^{-4} J_\lambda]^{1/6}$$

where k^2 is the directional relationship of transition dipoles, Q_D is the quantum yield of the donor chromophore, n is the refractive index of the medium, and J_λ is the spectral overlap of the donor and acceptor. In fact, the optical spectra of the two Cu nanoclusters were different between solution and crystalline phases, which might be attributed to the variation of their electronic structures in different states. These results could be inferred from the PL and UV-vis absorption spectra. As shown in Figure S25, the UV-vis absorption spectra of Cu₈ and Cu₁₀ nanoclusters in CH₂Cl₂ solution display no obvious absorption band (Figure S25A), while strong absorptions in the crystal state were detected (Figure S25B). In this context, we couldn't deduce the parameter of the refractive index of these cluster crystals from the solution state, and thus the Förster radius was incalculable. Furthermore, we calculated the FRET rate (k_{FRET}) using the DFT calculated coupling

strength by exploiting the Fermi's golden rule. The FRET rate in Table S5 is estimated by Fermi's golden rule. In addition, the FRET parameters are further estimated given that:

$$k_{\text{FRET}} = k_{\text{D}} \left(\frac{R_0}{r} \right)^6$$

where k_{D} is the donor's fluorescence decay rate in the absence of the acceptor and R_0 is the Förster radius. Then we can further relate V_{cp} and r as:

$$[V_{\text{cp}}]^2 = \frac{\hbar R_0^6 k_{\text{D}}}{\text{FCWD} 2\pi} r^{-6}.$$

The linear fitting of $[V_{\text{cp}}]^2$ versus r^{-6} is shown in Figure S26. PL decay study has revealed k_{D} to be 5.12×10^6 eV/s. The Franck-Condon factor weighted density of states (FCWD) was estimated to be 0.304 from the overlap of the normalized experimental spectra. Thus, the Förster radius R_0 was estimated to be 23.8 Å. Accordingly, the FRET efficiency (E) in the different molecular distance was also given in Table S5.

In the Supplementary Information:

Figure S25. UV-vis spectra of the Cu nanoclusters in (A) solution and (B) crystal states.

Figure S26. Square of electronic coupling strength ($[V_{\text{cp}}]^2$) with respect to -6 power of the distance (r^{-6}).

Table S5. DFT calculated excitation energies (E), oscillator strength (f), centroid of holes/electrons, coupling strength (V_{cp}) between S_1 and S_2 , and FRET rate constant (k_{FRET}) for the $Cu_8@Cu_{10}$ cocystal.

Distance (Å)	14.9	17.0	20.3	23.5	29.1	39.8	65.0
E_{S1} (eV)	3.02	3.02	3.02	3.02	3.02	3.02	3.02
f_{S1}	0.016	0.016	0.016	0.016	0.016	0.016	0.016
E_{S2} (eV)	3.13	3.14	3.15	3.15	3.15	3.15	3.15
f_{S2}	0.007	0.007	0.008	0.008	0.009	0.009	0.009
S_1 centroid of holes	(8.04, -0.03, 0.03)	(7.46, 6.93, 9.78)	(8.66, 8.01, 7.29)	(9.78, 9.05, 4.68)	(11.74, 10.92, 0.48)	(15.55, 14.56, -8.09)	(25.16, 23.63, 29.07)
S_1 centroid of electrons	(7.74, -0.04, -0.03)	(7.40, 6.87, 9.98)	(8.61, 7.99, 7.37)	(9.75, 9.08, 4.83)	(11.67, 10.89, 0.57)	(15.55, 14.53, -7.90)	(25.17, 23.61, 29.10)
S_2 centroid of holes	(-8.68, 0.17, -0.18)	(0.09, 0.29, 25.66)	(0.03, 0.28, 25.77)	(-0.01, 0.28, 25.81)	(-0.03, 0.16, 25.84)	(-0.02, 0.01, 25.99)	(0.02, 0.04, 26.13)
S_2 centroid of electrons	(-4.93, 4.69, -1.24)	(-0.32, 5.49, 23.55)	(-0.27, 4.88, 23.82)	(-0.20, 4.05, 24.17)	(-0.12, 2.58, 24.84)	(-0.05, 0.92, 25.60)	(0.00, 0.15, 25.93)
V_{cp} (meV)	0.162	0.140	0.074	0.053	0.031	0.013	0.003
k_{FRET} (eV/s)	7.59×10^7	5.73×10^7	1.57×10^7	8.20×10^6	2.88×10^6	4.78×10^5	2.43×10^4
E^*	94.3%	88.3%	72.2%	51.9%	23.0%	4.4%	0.2%

$$*E = \frac{1}{1+(r/R_0)^6}$$

4. Highlight Advantages over Previous Studies: The authors should dedicate a section to discussing the advantages of the FRET process based on copper nanoclusters compared to previously reported silver nanoclusters. This comparative analysis will underscore the uniqueness and advancements offered by the current research.

Response: We thank the reviewer for the insightful comment. The corresponding discussions have been added to the revised Manuscript. Thank you!

We revised the Manuscript into the following statements:

(Page 6) In previous works, although the FRET process was achieved by using silver nanoclusters such as Ag_{16} and Ag_{29} ^{35,36}, the precise dipole orientations, the favorable relative position for energy transfer, and the variation in the electronic structures of cluster molecules were hard to “see” directly in these cases. Besides, due to the various optical performances, for instance, the susceptible luminescence properties with different excitation source⁵¹⁻⁵³, the adjustable emission wavelength^{54,55}, and the multiple excited state^{50,56}, copper-based nanoclusters have been exploited as potential candidates to accomplish the FRET.

5. Cite Relevant Copper Nanocluster Literature: Strengthen the manuscript's foundation by citing more papers related to copper nanoclusters. Addressing these points will further solidify the manuscript's significance, making it suitable for publication in Nature Communications.

Response: We thank the reviewer for the valuable suggestion and supporting publication of this paper. As suggested, the relevant references were included, and the relevant discussions have been added to the revised Manuscript. Thank you!

We revised the Manuscript into the following statements:

(Page 6) Besides, due to the various optical performances, for instance, the susceptible luminescence properties with different excitation source⁵¹⁻⁵³, the adjustable emission wavelength^{54,55}, and the multiple excited state^{50,56}, copper-based nanoclusters have been exploited as potential candidates to accomplish the FRET. Recently, increasing research has focused on the cocrystallization of heterogeneous nanoclusters⁵⁷⁻⁶¹.

- 51 Yuan, P. et al. Hybrid Thermally Activated Nanocluster Fluorophores for X-ray Scintillators. *ACS Energy Lett.* **8**, 5088-5097 (2023).
- 52 Huang, R.-W. et al. Radioluminescent Cu–Au Metal Nanoclusters: Synthesis and Self-Assembly for Efficient X-ray Scintillation and Imaging. *J. Am. Chem. Soc.* **145**, 13816-13827 (2023).
- 53 Xu, C. et al. A High-Nuclearity Copper Sulfide Nanocluster [S-Cu₅₀] Featuring a Double-Shell Structure Configuration with Cu(II)/Cu(I) Valences. *J. Am. Chem. Soc.* **145**, 25673-25685 (2023).
- 54 Peng, S.-K., Yang, H., Luo, D., Ning, G.-H. & Li, D. A Highly NIR Emissive Cu₁₆Pd₁ Nanocluster. *Small* 2306863 (2023).
- 55 Ma, X.-H. et al. High-Efficiency Pure Blue Circularly Polarized Phosphorescence from Chiral N-Heterocyclic-Carbene-Stabilized Copper(I) Clusters. *J. Am. Chem. Soc.* **145**, 25874-25886 (2023).
- 56 Shi, Y.-e., Ma, J., Feng, A., Wang, Z. & Rogach, A. L. Aggregation-induced emission of copper nanoclusters. *Aggregate* **2**, e112, (2021).
- 57 Huang, J.-H., Liu, L.-Y., Wang, Z.-Y., Zang, S.-Q. & Mak, T. C. W. Modular Cocrystallization of Customized Carboranylthiolate-Protected Copper Nanoclusters *via* Host-Guest Interactions. *ACS Nano* **16**, 18789-18794 (2022).

Reviewer #2 (Remarks to the Author):

Authors have made substantive revisions based on the comments of the reviewers. The manuscript is now recommended for its acceptance for publication.

Response: We thank the reviewer for supporting publication of this paper.

Reviewer #3 (Remarks to the Author):

The authors have answered many questions satisfactorily and the manuscript has been improved. However, there are still some unanswered questions and errors that needs to be solved, and some important information is missing.

Response: We are grateful for this reviewer's feedbacks that have helped us to significantly improve this paper. Thank you!

1. There is still no explanation for the suppression of FRET at low temperatures. In new sentence "The emergence of the 537 nm signal was rational because the non-radiative FRET process was restricted at low temperatures that was similar to previously reported ones 55,56...", a reference is made to previous studies, where emission from similar systems is changed at low temperatures, and it is explained in terms of thermally activated delayed fluorescence or suppression of non-radiative pathways via freezing of vibrational degrees of freedom. For FRET, similar mechanism does not apply as it does not need any thermal excitation. An alternative explanation must be looked for. Perhaps one possibility is that at low temperature, the non-radiative relaxation rate of the Cu₈ cluster decreases sufficiently to make QY for emission large enough to be observed. This is in line with Fig S7. After all, the observed emission intensity depends on all competing rates. The authors should elaborate more on this and either give a plausible explanation of the phenomenon, or admit that they don't have an explanation on it.

Additionally, there is repetition in the text: "and thus the excited energy was released via radiative transition (e.g. PL) of Cu₈ and Cu₁₀ components, respectively, and then the energy released via radiative transition (e.g. PL) of Cu₈ and Cu₁₀ clusters, respectively."

Response: We thank the reviewer for the professional comment. We revised the relevant discussions, and the mistake has been corrected in the Manuscript. Thank you!

We revised the Manuscript into the following statements:

(Page 8) The emergence of the 537 nm signal at low temperature was rational and might be due to the following two reasons: i) the non-radiative process in the cocrystallization system was restricted, and ii) the radiative transition (e.g., PL) of Cu₈ was strengthened, which enhanced the PL QY sufficiently and the corresponding emission could be observed; indeed, the emerged 537 nm signal was similar to the emission of the monocomponent Cu₈ nanocluster at low temperature^{62,63}.

2. I would still recommend to perform a Förster theory calculation of the energy transfer rate using the available data. As the spectra don't change radically from solution to crystalline phase, the states should be reasonably similar and the calculation should give a correct order-of-magnitude estimate, thus making an important contribution to the paper. However, as there is new computational data on energy transfer rate, I would not consider this critical.

Response: We thank the reviewer for the professional comment. The key issue in solving the FRET efficiency is knowing the Förster radius (R_0). In fact, the optical spectra of the two Cu nanoclusters were inconsistent between solution and crystalline phases, which might be attributed to their different electric structures in the two states. These results could be inferred from the PL and UV-vis absorption spectra. As shown in Figure R3, the UV-vis absorption spectra of Cu₈ and Cu₁₀ nanoclusters in CH₂Cl₂ solution display no obvious absorption band (Figure R3A), while strong

absorptions in the crystal state were detected (Figure R3B). In this context, we couldn't deduce the parameter of the refractive index of these cluster crystals from the solution state, and thus the Förster radius was incalculable *via* the following equation:

$$R_0 = 9.78 \times 10^3 [k^2 Q_D n^{-4} J_\lambda]^{1/6}$$

where k^2 is the directional relationship of transition dipoles, Q_D is the quantum yield of the donor chromophore, n is the refractive index of the medium, and J_λ is the spectral overlap of the donor and acceptor.

Figure R3. The UV-Vis spectra of these Cu nanoclusters in (A) solution and (B) crystal states.

We calculated the FRET rate (k_{FRET}) using the DFT calculated coupling strength by exploiting the Fermi's golden rule $k_{\text{FRET}} = \frac{2\pi}{\hbar} [V_{cp}]^2 \text{FCWD}$ (Table S5). In addition, the FRET parameters are

further estimated given that $k_{\text{FRET}} = k_D \left(\frac{R_0}{r}\right)^6$, where k_D is the donor's fluorescence decay rate in the absence of the acceptor and R_0 is the Förster radius. Then we can further relate V_{cp} and r as: $[V_{cp}]^2 = \frac{\hbar R_0^6 k_D}{\text{FCWD} 2\pi} r^{-6}$. The linear fitting of $[V_{cp}]^2$ versus r^{-6} is shown in Figure R4. PL

decay study has revealed k_D to be 5.12×10^6 eV/s. The Franck-Condon factor weighted density of states (FCWD) is estimated to be 0.304 from the overlap of the normalized experimental spectra. Thus, the Förster radius R_0 is estimated to be 23.8 Å. Accordingly, the FRET efficiency (E) in the different molecular distance was also given via the following equation:

$$E = \frac{1}{1 + (r/R_0)^6}$$

where R_0 is the Förster radius with a 50% transfer efficiency.

Figure R4. Square of electronic coupling strength ($[V_{cp}]^2$) with respect to the -6 power of the distance (r^{-6}).

We revised the Manuscript and the Supplementary Information into the following statements:

In the Manuscript:

(Page 11) Next, we tried to determine the Förster radius (R_0). R_0 satisfied the following equation:

$$R_0 = 9.78 \times 10^3 [k^2 Q_D n^{-4} J_\lambda]^{1/6}$$

where k^2 is the directional relationship of transition dipoles, Q_D is the quantum yield of the donor chromophore, n is the refractive index of the medium, and J_λ is the spectral overlap of the donor and acceptor. In fact, the optical spectra of the two Cu nanoclusters were different between solution and crystalline phases, which might be attributed to the variation of their electronic structures in different states. These results could be inferred from the PL and UV-vis absorption spectra. As shown in Figure S25, the UV-vis absorption spectra of Cu_8 and Cu_{10} nanoclusters in CH_2Cl_2 solution display no obvious absorption band (Figure S25A), while strong absorptions in the crystal state were detected (Figure S25B). In this context, we couldn't deduce the parameter of the refractive index of these cluster crystals from the solution state, and thus the Förster radius was incalculable. Furthermore, we calculated the FRET rate (k_{FRET}) using the DFT calculated coupling strength by exploiting the Fermi's golden rule. The FRET rate in Table S5 is estimated by Fermi's golden rule. In addition, the FRET parameters are further estimated given that:

$$k_{\text{FRET}} = k_D \left(\frac{R_0}{r} \right)^6$$

where k_D is the donor's fluorescence decay rate in the absence of the acceptor and R_0 is the Förster radius. Then we can further relate V_{CP} and r as:

$$[V_{\text{CP}}]^2 = \frac{\hbar R_0^6 k_D}{\text{FCWD} 2\pi} r^{-6}.$$

The linear fitting of $[V_{\text{CP}}]^2$ versus r^{-6} is shown in Figure S26. PL decay study has revealed k_D to be 5.12×10^6 eV/s. The Franck-Condon factor weighted density of states (FCWD) was estimated to be 0.304 from the overlap of the normalized experimental spectra. Thus, the Förster radius R_0 was estimated to be 23.8 Å. Accordingly, the FRET efficiency (E) in the different molecular distance was also given in Table S5.

In the Supplementary Information:

Figure S25. UV-vis spectra of the Cu nanoclusters in (A) solution and (B) crystal states.

Figure S26. Square of electronic coupling strength ($[V_{cp}]^2$) with respect to -6 power of the distance (r^{-6}).

Table S5. DFT calculated excitation energies (E), oscillator strength (f), centroid of holes/electrons, coupling strength (V_{cp}) between S_1 and S_2 , and FRET rate constant (k_{FRET}) for the $Cu_8@Cu_{10}$ cocrystal.

Distance (Å)	14.9	17.0	20.3	23.5	29.1	39.8	65.0
E_{S1} (eV)	3.02	3.02	3.02	3.02	3.02	3.02	3.02
f_{S1}	0.016	0.016	0.016	0.016	0.016	0.016	0.016
E_{S2} (eV)	3.13	3.14	3.15	3.15	3.15	3.15	3.15
f_{S2}	0.007	0.007	0.008	0.008	0.009	0.009	0.009
S_1 centroid of holes	(8.04, -0.03, 0.03)	(7.46, 6.93, 9.78)	(8.66, 8.01, 7.29)	(9.78, 9.05, 4.68)	(11.74, 10.92, 0.48)	(15.55, 14.56, -8.09)	(25.16, 23.63, 29.07)
S_1 centroid of electrons	(7.74, -0.04, -0.03)	(7.40, 6.87, 9.98)	(8.61, 7.99, 7.37)	(9.75, 9.08, 4.83)	(11.67, 10.89, 0.57)	(15.55, 14.53, -7.90)	(25.17, 23.61, 29.10)
S_2 centroid of holes	(-8.68, 0.17, -0.18)	(0.09, 0.29, 25.66)	(0.03, 0.28, 25.77)	(-0.01, 0.28, 25.81)	(-0.03, 0.16, 25.84)	(-0.02, 0.01, 25.99)	(0.02, 0.04, 26.13)
S_2 centroid of electrons	(-4.93, 4.69, -1.24)	(-0.32, 5.49, 23.55)	(-0.27, 4.88, 23.82)	(-0.20, 4.05, 24.17)	(-0.12, 2.58, 24.84)	(-0.05, 0.92, 25.60)	(0.00, 0.15, 25.93)
V_{cp} (meV)	0.162	0.140	0.074	0.053	0.031	0.013	0.003
k_{FRET} (eV/s)	7.59×10^7	5.73×10^7	1.57×10^7	8.20×10^6	2.88×10^6	4.78×10^5	2.43×10^4
E^*	94.3%	88.3%	72.2%	51.9%	23.0%	4.4%	0.2%

$$*E = \frac{1}{1+(r/R_0)^6}$$

3. I have a comment on the answer of the authors to the comment 4, regarding the possibility of the Dexter mechanism. The authors state that “the center-to-center distance of the transition dipole of these two Cu nanoclusters was identified as about 15-20 angstroms, much longer than that of the Dexter energy transfer. Besides, the spatial locations of Cu8 and Cu10 nanoclusters in the cocrystal were fixed, demonstrating that the intermolecular collision was impossible. In this

context, the energy transfer between the two Cu nanoclusters is more inclined to FRET than Dexter energy transfer.”

However, It is not true that intermolecular collisions are not possible in a crystal. Near neighbors undergo thermally activated collisions, which modulate for example the Dexter energy transfer rate.

Additionally, For Dexter mechanism, the center to center distance of the transition dipoles is not a proper parameter. Dexter mechanism requires overlap of electronic orbitals. Thus, the relevant parameter is the distance compared to the Van der Waals distance (see, for example the Turro book). The authors may want to once more consider the possibility of the Dexter mechanism in this light.

Response: We thank the reviewer for the insightful suggestion. We confirmed the overlapping of the electronic orbitals of the two Cu nanoclusters in the cocrystal lattice by DFT calculations. As the results are shown in Figure R5, the FRET rate is correlated with the transition dipole-dipole coupling strength. While Dexter energy transfer is the direct electron exchange process that requires the wavefunction overlap of HOMO (or LUMO) at the donor and acceptor. The spatial distributions of HOMO and LUMO at Cu₈ (donor) and Cu₁₀ (acceptor) are shown in the Figure R6. The minimum distance between donor and acceptor LUMO (HOMO) is 10.7 (11.9) Å. Meanwhile, the overlap of donor and acceptor LUMO (HOMO) is negligible, indicating that the direct electron (hole) transfer is prohibited by the poor wavefunction overlap. Thus, the Dexter energy transfer is less favored in Cu₈@Cu₁₀ cocrystal. The relevant discussions have been added to the revised Manuscript and Supplementary Information. Thank you!

Figure R5. Schematic process of Förster resonance energy transfer and Dexter energy transfer.

Figure R6. Spatial distribution of HOMO and LUMO on donor (Cu_8) and acceptor (Cu_{10}).

We revised the Manuscript into the following statements:

(Page 10) We also considered the possibility of Dexter energy transfer between Cu_8 and Cu_{10} nanoclusters. Dexter energy transfer is the direct electron exchange process that requires the wavefunction overlap of HOMO (or LUMO) at the donor and acceptor, while the FRET rate is correlated with the transition dipole-dipole coupling strength (Figure S22). The spatial distribution of HOMO and LUMO at Cu_8 (donor) and Cu_{10} (acceptor) is shown in Figure S23. The minimum distance between donor and acceptor LUMO (HOMO) is 10.7 (11.9) Å. Meanwhile, the overlap of donor and acceptor LUMO (HOMO) is negligible, indicating that the direct electron (hole) transfer is prohibited by the poor wavefunction overlap. Thus, the Dexter energy transfer is less favored in the $\text{Cu}_8@ \text{Cu}_{10}$ cocrystal.

4. The calculations suggest that the energy transfer occurs from the S2 state of the Cu_8 cluster. In this case, isn't the non-radiative relaxation from S2 to S1, which should be very rapid, a problem? What is the rate for this relaxation? There should be information on the properties of the S2 state and a comment on this question.

Response: We thank the reviewer for the professional comments. FRET is the energy transfer mechanism between donor and acceptor molecules. The donor (Cu₈), initially pumped to its electronic excited state (S₂), may transfer energy to excite the acceptor (Cu₁₀) to its excited state (S₁) through non-radiative coupling. The non-radiative Coulombic interaction dipole-dipole between S₁ and S₂ corresponded to the FRET in the Cu₈@Cu₁₀ cocrystal. The coupling strength and estimated FRET rate are shown in Figure 4B and Table S5. In comparison, we investigated the direct radiative transition by evaluating the electric transition dipole moments $\langle i|-r|j\rangle$ and its oscillator strength (Table S6). The S₁→S₂ oscillator strength is only 3.00×10⁻⁵. The low oscillator strength indicated a slow radiative transition rate. Therefore, the non-radiative FRET is the favored energy transfer mechanism between Cu₈ and Cu₁₀ in their cocrystal system. The corresponding discussions have been added in the revised Manuscript and Supplementary Information. Thank you!

We revised the Manuscript and the Supplementary Information into the following statements.

In the Manuscript:

(Page 10) FRET is the energy transfer mechanism between donor and acceptor molecules. The donor (Cu₈), initially pumped to its electronic excited state (S₂), may transfer energy to excite the acceptor (Cu₁₀) to its excited state (S₁) through non-radiative coupling. The non-radiative Coulombic interaction dipole-dipole between S₁ and S₂ corresponded to the FRET in the Cu₈@Cu₁₀ cocrystal. The coupling strength and estimated FRET rate are shown in Figure 4B and Table S5. In comparison, we investigated the direct radiative transition by evaluating the electric transition dipole moments $\langle i|-r|j\rangle$ and its oscillator strength (Table S6). The S₁→S₂ oscillator strength was only 3.00×10⁻⁵. The low oscillator strength indicated a slow radiative transition rate. Therefore, the non-radiative FRET is the favored energy transfer mechanism between Cu₈ and Cu₁₀.

In the Supplementary Information:

Table S6. Electric transition dipole moments and its oscillator strength among S₀, S₁, S₂, and S₃ for Cu₈@Cu₁₀ cluster.

Initial State	Final State	Transition Dipole (a.u.)			Energy Difference (eV)	Oscillator Strength
		X	Y	Z		
S0	S1	0.3835	-0.6109	0.0030	3.0158	1.61×10 ⁻²
S0	S2	0.3500	0.0938	-0.0849	3.1327	6.90×10 ⁻³
S0	S3	0.2702	0.0685	-0.0528	3.1531	3.70×10 ⁻³
S1	S2	-0.0992	-0.0138	0.0009	0.1169	3.00×10 ⁻⁵
S1	S3	-0.0242	0.0024	0.0000	0.1373	0.0
S2	S3	1.6942	1.9572	-0.4331	0.0204	3.44×10 ⁻³

5. There are inconsistencies regarding the electronic coupling strength. In the rebuttal letter, the coupling is given in eV as well as in the Fig 4E of the main article, whereas in text and Fig. 4B, it is given in meV. These inconsistencies should be corrected. I suppose meV is correct.

Response: We thank the reviewer for the helpful comment. The unit should be meV. This mistake has been corrected in the revised Figure 4. Thank you!

6. The fitting data on Fig. S6 seems strange. First of all, there are two almost equal lifetime components (0.4 and 0.37 microseconds). The small difference indicates that three components

should be sufficient. The authors should try that. Second, it is not clear how the average lifetime of 0.47 microseconds was obtained. Is it an amplitude weighted average of all the lifetimes? This should be explained. In any case, the average lifetime seems quite low. In Fig. S6, the scale of the y-axis should be included (also in S9). Then it will be possible to judge if the average lifetime reflects the time evolution correctly.

Additionally, what is the instrumental response time? Is it possible that the shortest lifetime component actually is giving the instrumental response time?

Response: We thank the reviewer for the professional comments. All the lifetimes were amplitude-weighted average and simulated by *DAS 6.0 Analysis* software. As suggested, we re-simulated the PL lifetime of Cu₈ nanoclusters. The results of the lifetime calculated by the 3-order exponential fitting showed a large CHISQ value, indicating that the fitting result is poor with large differences (Figure R7). While the lifetime of the Cu₈ cluster calculated by the 4-order exponential fitting showed a more credible value (CHISQ = 1.18). The detailed report of the PL lifetime has been added to the Supplementary Information as the Attachment S1 to S3. Thank you!

Figure R7. The PL lifetime of Cu₈ nanoclusters simulated by 3-order exponential fitting.

The Laser diode heads (DD-375L) was used as excitation light source in the PL lifetime experiments (https://static.horiba.com/fileadmin/Horiba/Products/Scientific/Optical_Components_and_OEM/DeltaDiode/DeltaDiode_Brochure.pdf). The maximum pulse width was 70 ps, remarkably less than the PL lifetime of the cluster samples in this work. In this context, the pulse width of the instrumental could be ignored.

We revised the Supplementary Information into the following statements:

Figure S6. The emission lifetime of the Cu₈ nanocluster at room temperature.

7. In Figs. S9 and S15, there seems to be a rise time component. Is that correct. It should be mentioned and its significance briefly discussed.

Response: We thank the reviewer for the professional comment. According to the simulation results by the *DAS 6.0 Analysis* software, the average lifetime of the cocrystal was increased compared with the monocomponent Cu₁₀ nanocluster. We deduced that the cocrystallized system has undergone longer energy transfer processes including the excitation process of the Cu₈ nanocluster, the FRET process, and the energy release process of the Cu₁₀ nanocluster. We updated the relevant discussions in the revised Manuscript. Thank you!

We revised the Manuscript into the following statements:

(Page 11) Therefore, the longer average PL lifetime of the cocrystallized Cu₈@Cu₁₀ than the monocomponent Cu₈ or Cu₁₀ might be attributed to it undergoing overall energy transfer processes including the excitation process of the Cu₈ nanocluster, the FRET process, and the energy release process of the Cu₁₀ nanocluster.

8. In the introduction, the statement: "Herein, the FRET was achieved for the first time between nanoclusters at the atomic level by exploiting..." is not justified. I think there are previous studies of FRET between atomically precise nanoclusters. The sentence should be modified.

Response: We thank the reviewer for insightful comments. Although previous studies have accomplished the FRET with the participation of atomically precise nanoclusters, these cases were almost achieved in solutions. In this work, we realized the FRET between nanoclusters at the atomic level *via* cocrystallization-induced spatial confinement strategy; that is, the FRET was accomplished in the crystal system. We have re-emphasized the strategy in the revised Manuscript. Thank you!

We revised the Manuscript into the following statements:

(Page 3) Herein, the FRET was achieved for the first time between nanoclusters at the atomic level by exploiting the cocrystallization-induced spatial confinement between two fluorescent copper clusters.

9. For the Cu₈ cluster, the emission QY is stated as 4.2 %. Is that measured at room temperature? This should be given. The temperature dependence seems to be very strong especially between 200 and 230 K (Fig. S8). It would be interesting to see the T-dependence plotted. Is there an Arrhenius-type of behavior?

Although it is very hard to see from Fig. S7, it seems that the photoluminescence intensity increases by a factor much larger than 20, perhaps by 100, when going from the room temperature to 80 K and presumably it would increase further at lower temperatures. Referring to point 9, if the QY is 4.2 % at RT, the large increase of intensity would mean >100 % QY at low T. This possible discrepancy should be considered and explained.

Response: We thank the reviewer for the professional comment. All the absolute PL QY experiments were measured at room temperature, and it has been updated in the revised Supplementary Information.

According to the *Arrhenius Law*:

$$\ln k = -\frac{E_a}{RT} + \ln A$$

where the k is the radiative rate, which is in direct proportion to the PL intensity. The PL spectra at 210 K and 220 K were added to the temperature-dependent PL spectra of the Cu_8 nanocluster. The relationship between the $\ln k$ and $1/T$ is given in Figure R8.

Figure R8. The relationship between the $\ln k$ and $1/T$.

The results indicated that it might be the Arrhenius-type of behavior in some temperature intervals such as 80 K to 200 K, 200 K to 230 K, and 230 K to 290 K.

Although the PL changed tremendously in intensity, it does not mean that the QY changes with a similar tendency during the temperature decrease. The QY value was also related to the absorption of the samples. We further measured the excitation spectra of the Cu_8 nanocluster to illustrate the variation of absorption intensity, and the results demonstrated that absorption also has a huge increase in intensity. Therefore, the variation of QY value can't be comprehended with the change in PL intensity. We tried to estimate the relative quantum yield at 80 K by using the integral area of the PL spectra and the excitation spectra intensity at different temperatures with the following equation:

$$Q_F = Q_R \frac{I_F A_R}{I_R A_F}$$

where Q_F is the quantum yield of the unknown fluorescent sample, Q_R is the quantum yield of the reference standard, I_F and I_R are the integrated fluorescence intensities for the sample and the reference, respectively, and A_F and A_R are the excitation spectra intensity of the sample and reference, respectively. Thus, the relative quantum yield of the Cu_8 nanocluster at 80 K was $\sim 14.6\%$, much less than 100%. For clarity, we added the relevant discussions into the revised Manuscript and Supplementary Information. Thank you!

We revised the Manuscript and Supplementary Information into the following statements:

In the Manuscript:

(Page 5) Although the PL changed tremendously in intensity of the Cu_8 nanocluster, the PL QY did not follow the same changing trend. Indeed, the QY value was also related to the absorption of the

cluster sample. The excitation spectra of Cu₈ were measured to illustrate the variation of absorption intensity, and the results demonstrated that the absorption of Cu₈ also displayed a remarkable enhancement in intensity with the temperature decreasing (Figure S7B).

In the Supplementary Information:

(Page 2) The absolute PL QY test was carried out with integrating sphere at room temperature and calculated using the *FluorEssence* software.

Figure S7. Temperature-dependent (A) PL and (B) excitation spectra of the Cu₈ nanocluster. (C) The relationship between $\ln k$ and $1/T$.

According to the *Arrhenius Law*:

$$\ln k = -\frac{E_a}{RT} + \ln A$$

where k is the radiative rate, which is in direct proportion to the PL intensity. The relationship between the $\ln k$ and $1/T$ is given in Figure S7C. The results indicated that it might be the Arrhenius-type of behavior in some temperature intervals such as 80 K to 200 K, 200 K to 230 K, and 230 K to 290 K.

We tried to estimate the relative quantum yield at 80 K by using the integral area of the PL spectra and the excitation spectra intensity at different temperatures with the following equation:

$$Q_F = Q_R \frac{I_F A_R}{I_R A_F}$$

where Q_F is the quantum yield of the unknown fluorescent sample, Q_R is the quantum yield of the reference standard, I_F and I_R are the integrated fluorescence intensities for the sample and the reference, respectively, and A_F and A_R are the excitation spectra intensity of the sample and reference, respectively. Thus, the relative quantum yield of the Cu₈ nanocluster at 80 K was ~ 14.6%, much less than 100%. Similarly, the relative quantum yields of Cu₁₀ and Cu₈@Cu₁₀ nanoclusters were also less than 100% at 80 K.

10. In Fig. 4B, it would be interesting to see if the calculated energy transfer rate follows $1/r^6$ distance-dependence, as it should. Perhaps a fit could be tried and possibly included in the Figure.

Response: We thank the reviewer for the professional comment. Because k_{FRET} is proportional to the square of the electronic coupling strength ($[V_{cp}]^2$), we fitted $[V_{cp}]^2$ with respect to r^{-6} . The linear fitting in Figure S26 verified the r^{-6} distance dependence of k_{FRET} .

We revised the Supplementary Information into the following statements:

Figure S26. Square of electronic coupling strength ($[V_{cp}]^2$) with respect to -6 power of the distance (r^{-6}).

11. The interpretation of the amorphization experiment (observation of dual emission) of the co-crystal does not sound right. It seems highly unlikely that the orientation factor could explain the suppression of energy transfer. I suppose a better explanation is that the individual clusters are again segregating into separate phases. In an amorphous crystal, there would in any case always exist favorable directions for energy transfer between some pairs of nearest neighbor clusters.

Response: We thank the reviewer for the insightful comment. As suggested, we revised the relevant discussion in the Manuscript. Thank you!

We revised the Manuscript into the following statements:

(Page 12) In this powder, the intermolecular distance and the dipole orientation of Cu_8 and Cu_{10} nanoclusters were uncontrolled. Besides, the two copper nanoclusters might segregate into single separate crystal phases.

12. When describing the lifetime fitting results, instead of using terminology “two compositions” it would be better to use “two lifetime components”.

Response: We thank the reviewer for the professional comment. As suggested, we have revised the relevant content in the Manuscript. Thank you!

Finally, we thank all reviewers for their helpful suggestions and comments.

REVIEWER COMMENTS

Reviewer #1 (Remarks to the Author):

Authors have done substantial amount work to address the reviewer concern. With the new results manuscript quality has improved to very quality. I recommend the manuscript should be published in the current format.

Reviewer #3 (Remarks to the Author):

The authors have answered some questions satisfactorily and the manuscript has been improved. However, there are still some problems that need to be solved. I elaborate below.

1. On page 5, the treatment of the emission lifetimes is still not satisfactory. The authors rejected my previous suggestion to use only 3 lifetime components in the fitting. I did not fully understand the point when they showed in the answer three different fits and concluded that the ChiSq values are higher with three components than with four components. The values in the three fits were 0.69, 1.43 and 0.73, while for the four-component fit it is 1.18. What are the three different fits, is not clear to me.

Nevertheless, now when the authors have added an attachment S1, which shows the fitting parameters I see that the component, which has a lifetime of 0.4 microseconds has a negative amplitude. Thus, it has no physical meaning and it would be better to either use three components or restrict the amplitudes to positive values.

2. In the end of page 5, there is new text on the temperature-dependence of the emission and absorption. In my previous report, I asked about the emission QY, which seemed to go above 100 % at low temperatures. The authors opposed this by claiming that the absorption also changes with temperature. However, they are backing up this claim with the temperature-dependence of the excitation spectrum, which is not a correct procedure. Excitation spectrum is a product of absorption and emission QY and thus one cannot deduce from it independently the change in absorption intensity. In fact, looking at Fig. S7 A and B, it seems that the temperature-dependence of emission and excitation is pretty much the same. This would mean that the absorption strength does not change and my original question is still valid. The authors could plot the normalized emission intensity and excitation spectrum intensity in the same figure and verify if my doubt is correct. If yes, the original question should still be answered.

3. On page 10, the calculations on the FRET process are discussed. A general problem here is that it is confusing to use the S1, S2 terminology for the supersystem, consisting of both clusters.

Conventionally, S1 refers to the first singlet excited state of a single molecule, S2 to its second excited state etc. The authors use S1 for the first excited state of the supersystem consisting of both clusters. Thus, S2 refers to the first excited state of the acceptor cluster. I find this terminology very confusing and I would propose to use an alternative naming. One could use for example S1D, S2D for donor states and S1A, S2A etc. for the acceptor states.

There is also one more problem in these calculations. The authors have calculated oscillator strength for the S1->S2 transition. That would correspond to radiative transition from the first excited state of the acceptor to the first excited state of the donor, which does not make sense. Such a transition is not a radiative electronic transition of a single quantum system but would involve electron exchange between acceptor and donor. This part should be revised. The same problem concerns also other transitions in S6.

4. On page 12 and in Table S5, the rate constants k_D and k_{FRET} are reported in units of eV/s. This is not correct. The unimolecular rate constant should be in units of s⁻¹.

5. When describing the lifetime fitting results, instead of using terminology "two compositions" it would be better to use "two lifetime components". I pointed out this already in the previous review report, but the problem still persists in many places, such as pages 6 and 8.

We thank all reviewers for their helpful comments. The point-by-point responses are shown in blue in this letter, and revisions are in red. Besides, the revisions in the revised manuscript are highlighted.

Reviewer #1 (Remarks to the Author):

Authors have done substantial amount work to address the reviewer concern. With the new results manuscript quality has improved to very quality. I recommend the manuscript should be published in the current format.

Response: We thank the reviewer for supporting publication of this paper.

Reviewer #3 (Remarks to the Author):

The authors have answered some questions satisfactorily and the manuscript has been improved. However, there are still some problems that need to be solved. I elaborate below.

Response: We thank the reviewer for the professional and valuable suggestions to remarkably improve the Manuscript. Thank you!

1. On page 5, the treatment of the emission lifetimes is still not satisfactory. The authors rejected my previous suggestion to use only 3 lifetime components in the fitting. I did not fully understand the point when they showed in the answer three different fits and concluded that the ChiSq values are higher with three components than with four components. The values in the three fits were 0.69, 1.43 and 0.73, while for the four-component fit it is 1.18. What are the three different fits, is not clear to me.

Nevertheless, now when the authors have added an attachment S1, which shows the fitting parameters I see that the component, which has a lifetime of 0.4 microseconds has a negative amplitude. Thus, it has no physical meaning and it would be better to either use three components or restrict the amplitudes to positive values.

Response: We thank the reviewer for the professional comments. CHISQ (chi-square test) is a test for independence given by the *DAS 6.0 Analysis* software. The CHISQ value that is close to 1 means its well-fitting with the experimental results. As suggested, we reattempted the simulation of the PL lifetime of the Cu₈ nanocluster fitted with a 3-order exponential fitting equation by *DAS 6.0 Analysis* software, and the average emission lifetime was 1.20 μs with three-lifetime components ($\tau_1 = 4.12 \mu\text{s}$, $\tau_2 = 14.2 \mu\text{s}$, and $\tau_3 = 0.21 \mu\text{s}$; please see the Figure S6). The difference in the average PL lifetime from the previous version (0.47 μs) of the Cu₈ nanocluster was due to the different fitting methods. The corresponding discussions have been revised in the Manuscript and Supplementary Information as suggested. Thank you!

We revised the Manuscript and the Supplementary Information into the following statements:

In the Manuscript:

(Page 5) the average emission lifetime was 1.20 μs with three lifetime components ($\tau_1 = 4.12 \mu\text{s}$, $\tau_2 = 14.2 \mu\text{s}$, and $\tau_3 = 0.21 \mu\text{s}$; Figure S6)...

Figure 3F. Emission lifetime of cocrystallized Cu₈ and Cu₁₀ clusters at room temperature.

In the Supplementary Information:

(Page 2) The PL lifetime of the Cu₈ crystal was complicated and was calculated by a 3-order exponential fitting equation: $A + B_1 \cdot \exp(-i/T_1) + \dots + B_3 \cdot \exp(-i/T_3)$.

Figure S6. The emission lifetime of the Cu₈ nanocluster at room temperature.

Table S4. Photophysical data of Cu₈, Cu₁₀, and Cu₈@Cu₁₀ cluster.

Cluster	$\lambda_{em}(nm)$	$\tau_{av}(\mu s)$	Φ_{em}	$\kappa_r (s^{-1})$	$\kappa_{nr} (s^{-1})$
Cu ₈	515	1.20	~ 4.2 %	3.50×10^4	7.98×10^6
Cu ₁₀	655	5.74	~ 41.1 %	7.16×10^4	1.03×10^5
Cu ₈ @Cu ₁₀	640	6.54	~ 43.3 %	6.62×10^4	8.67×10^4

2. In the end of page 5, there is new text on the temperature-dependence of the emission and absorption. In my previous report, I asked about the emission QY, which seemed to go above 100 % at low temperatures. The authors opposed this by claiming that the absorption also changes with temperature. However, they are backing up this claim with the temperature-dependence of the excitation spectrum, which is not a correct procedure. Excitation spectrum is a product of absorption and emission QY and thus one cannot deduce from it independently the change in absorption intensity. In fact, looking at Fig. S7 A and B, it seems that the temperature-dependence of emission and excitation is pretty much the same. This would mean that the absorption strength does not change and my original question is still valid. The authors could plot the normalized emission intensity and excitation spectrum intensity in the same figure and verify if my doubt is correct. If yes, the original question should still be answered.

Response: We thank the reviewer for the professional comments. After several efforts, the absorption spectra of these Cu nanoclusters at 80 and 290 K were monitored. For better analyzing the PL QY of the Cu₈ nanocluster at 80 K, its absorption spectrum at 80 K was carried out to evaluate the relative quantum yield with the following equation:

$$Q_{80 K} = Q_{290 K} \frac{I_{80 K} A_{290 K}}{I_{290 K} A_{80 K}}$$

where $Q_{80 K}$ is the quantum yield at 80 K, $Q_{290 K}$ is the quantum yield at 290 K, $I_{80 K}$ and $I_{290 K}$ are the integrated fluorescence intensity at 80 and 290 K, respectively, and $A_{80 K}$ and $A_{290 K}$ are the absorption spectra intensity at 80 and 290 K, respectively.

As shown in Figure R1, the integrated fluorescence intensity exhibited a 53.4-time enhancement from 290 to 80 K, and the absorption intensity showed a 2.47-time enhancement. The relative

quantum yield of the Cu₈ nanocluster at 80 K was determined as ~ 88.5 %, not exceeding 100 %. In addition, the relative quantum yields of the Cu₁₀ and Cu₈@Cu₁₀ at 80 K were ~ 53.5 % and ~ 60.4 %, respectively. The related results have been updated in the revised Manuscript and Supplementary Information. Thank you!

Figure R1. The (A) emission and (B) absorption spectra of Cu₈ nanocluster at 80 K and 290 K.

We revised the Manuscript and the Supplementary Information into the following statements:

In the Manuscript:

(Page 5) Although the PL changed tremendously in intensity of the Cu₈ nanocluster (about 54 times; Figure S7C), the PL QY did not follow the same changing trend. Indeed, the QY value was also related to the absorption of the cluster sample. In this context, the absorption spectrum of Cu₈ at 80 K was measured to determine its relative QY at low temperatures. The results demonstrated that the absorption of Cu₈ displayed a 2.47-time enhancement in intensity with the temperature decreasing (Figure S7D). Accordingly, the relative QY of the Cu₈ nanocluster was ~ 88.5 % at 80 K.

In the Supplementary Information:

(Page 2) **UV-Vis absorption spectra:** The UV-Vis absorption spectra in the solution state were collected on a PerkinElmer Lambda 465 spectrophotometer. The UV-Vis absorption spectra in the solid state were carried on a Shimadzu 3600-plus spectrophotometer with an integrating sphere.

Figure S7. (A) Temperature-dependent PL spectra of the Cu₈ nanocluster. (B) The relationship between the $\ln k$ and $1/T$. The comparison of (C) integrated fluorescence intensity and (D) absorption intensity of the Cu₈ nanocluster at 290 and 80 K, respectively.

(Page 6) We tried to estimate the relative quantum yield at 80 K of Cu₈ nanocluster by using the integral area of the PL spectra and the absorption spectra intensity at different temperatures with the following equation:

$$Q_{80 K} = Q_{290 K} \frac{I_{80 K} A_{290 K}}{I_{290 K} A_{80 K}}$$

where $Q_{80 K}$ is the quantum yield at 80 K, $Q_{290 K}$ is the quantum yield at 290 K, $I_{80 K}$ and $I_{290 K}$ are the integrated fluorescence intensity at 80 and 290 K, respectively, and $A_{80 K}$ and $A_{290 K}$ are the absorption spectra intensity at 80 and 290 K, respectively. Thus, the relative quantum yield of the Cu₈ nanocluster at 80 K was $\sim 88.5\%$, not exceeding 100%. Similarly, the relative quantum yields of Cu₁₀ and Cu₈@Cu₁₀ nanoclusters were also less than 100% at 80 K.

Figure S10. (A) Temperature-dependent PL spectra of Cu₁₀ nanocluster. The comparison of (B) integrated fluorescence intensity and (C) absorption intensity of the Cu₁₀ nanocluster at 290 and 80 K, respectively. The relative quantum yield of the Cu₁₀ nanocluster at 80 K was given as $\sim 53.5\%$.

Figure S17. The comparison of (A) integrated fluorescence intensity and (B) absorption intensity of the Cu₈@Cu₁₀ co-crystal at 290 and 80 K, respectively. The relative quantum yield of the Cu₈@Cu₁₀ co-crystal at 80 K was given as ~ 60.4 %.

3. On page 10, the calculations on the FRET process are discussed. A general problem here is that it is confusing to use the S1, S2 terminology for the supersystem, consisting of both clusters. Conventionally, S1 refers to the first singlet excited state of a single molecule, S2 to its second excited state etc. The authors use S1 for the first excited state of the supersystem consisting of both clusters. Thus, S2 refers to the first excited state of the acceptor cluster. I find this terminology very confusing and I would propose to use an alternative naming. One could use for example S1D, S2D for donor states and S1A, S2A etc. for the acceptor states.

There is also one more problem in these calculations. The authors have calculated oscillator strength for the S1->S2 transition. That would correspond to radiative transition from the first excited state of the acceptor to the first excited state of the donor, which does not make sense. Such a transition is not a radiative electronic transition of a single quantum system but would involve electron exchange between acceptor and donor. This part should be revised. The same problem concerns also other transitions in S6.

Response: We appreciate the reviewer for highlighting the confusing usage of terminology. We agree with the reviewer that using S_{1,D} and S_{1,A} would enhance the clarity in our statements. We have revised and highlighted the terminology accordingly in the Manuscript and Supplementary Information. Thank you!

The calculation of the oscillator strength for the S1->S2 transition is conducted to simply compare the non-radiative and radiative processes to verify the nature of the energy transfer. The low oscillator strength suggests that the radiative process is less favored. In contrast, the coupling strength and estimated FRET rate are illustrated in Figure 4B and Table S5. The significant coupling strength indicates that non-radiative energy transfer (FRET) is favored.

We revised the Manuscript and the Supplementary Information into the following statements:

In the Manuscript:

(Page 10) In the Cu₈@Cu₁₀ cocrystal, as shown in Figure S20, the MLCT-corresponded excited state (S_{1,A}) localized on the Cu₁₀ cluster molecule (HOMO-1 to LUMO), while the excited state (S_{1,D}) localized on the Cu₈ cluster molecule (HOMO to LUMO+1).

In the Supplementary Information:

Figure S21. DFT calculated (A) relative energies of frontier orbitals in Cu₈@Cu₁₀ cocrystal with the energy levels in units of eV and the spatial distribution of electrons/holes for (B) S_{1,D} on Cu₈ and (C) S_{1,A} on Cu₁₀ with isosurfaces at 0.02 a.u.

Table S6. electric transition dipole moments and its oscillator strength among S₀, S_{1,A}, and S_{2,D} for Cu₈@Cu₁₀ cluster.

Initial State	Final State	Transition Dipole (a.u.)			Energy Difference (eV)	Oscillator Strength
		X	Y	Z		
S ₀	S _{1,A}	0.3835	-0.6109	0.0030	3.0158	1.61×10 ⁻²
S ₀	S _{1,D}	0.3500	0.0938	-0.0849	3.1327	6.90×10 ⁻³
S _{1,A}	S _{1,D}	-0.0992	-0.0138	0.0009	0.1169	3.00×10 ⁻⁵

4. On page 12 and in Table S5, the rate constants k_D and k_{FRET} are reported in units of eV/s. This is not correct. The unimolecular rate constant should be in units of s⁻¹.

Response: We thank the reviewer for the professional comments. the relevant discussions have been updated in the revised Manuscript and Supplementary Information. Thank you!

We revised the Manuscript and the Supplementary Information into the following statements:

In the Manuscript:

(Page 12) PL decay study has revealed k_D to be $2.01 \times 10^6 \text{ s}^{-1}$.

In the Supplementary Information:

Table S5. DFT calculated excitation energies (E), oscillator strength (f), centroid of holes/electrons, coupling strength (V_{cp}) between S₁ and S₂, and FRET rate constant (k_{FRET}) for the Cu₈@Cu₁₀ cocrystal.

Distance (Å)	14.9	17.0	20.3	23.5	29.1	39.8	65.0
E_{S1} (eV)	3.02	3.02	3.02	3.02	3.02	3.02	3.02
f_{S1}	0.016	0.016	0.016	0.016	0.016	0.016	0.016
E_{S2} (eV)	3.13	3.14	3.15	3.15	3.15	3.15	3.15
f_{S2}	0.007	0.007	0.008	0.008	0.009	0.009	0.009

S₁ centroid of holes
 (8.04, -0.03, 0.03) (7.46, 6.93, 9.78) (8.66, 8.01, 7.29) (9.78, 9.05, 4.68) (11.74, 10.92, 0.48) (15.55, 14.56, -8.09) (25.16, 23.63, 29.07)

S₁ centroid of electrons	(7.74, -0.04, -0.03)	(7.40, 6.87, 9.98)	(8.61, 7.99, 7.37)	(9.75, 9.08, 4.83)	(11.67, 10.89, 0.57)	(15.55, 14.53, -7.90)	(25.17, 23.61, 29.10)
S₂ centroid of holes	(-8.68, 0.17, -0.18)	(0.09, 0.29, 25.66)	(0.03, 0.28, 25.77)	(-0.01, 0.28, 25.81)	(-0.03, 0.16, 25.84)	(-0.02, 0.01, 25.99)	(0.02, 0.04, 26.13)
S₂ centroid of electrons	(-4.93, 4.69, -1.24)	(-0.32, 5.49, 23.55)	(-0.27, 4.88, 23.82)	(-0.20, 4.05, 24.17)	(-0.12, 2.58, 24.84)	(-0.05, 0.92, 25.60)	(0.00, 0.15, 25.93)
V_{cp} (meV)	0.162	0.140	0.074	0.053	0.031	0.013	0.003
k_{FRET} (s⁻¹)	7.59×10 ⁷	5.73×10 ⁷	1.57×10 ⁷	8.20×10 ⁶	2.88×10 ⁶	4.78×10 ⁵	2.43×10 ⁴
E*	97.7%	95.1%	87.1%	73.7%	43.7%	10.6%	0.6%

$$*E = \frac{1}{1+(r/R_0)^6}$$

5. When describing the lifetime fitting results, instead of using the terminology “two compositions” it would be better to use “two lifetime components”. I pointed out this already in the previous review report, but the problem still persists in many places, such as pages 6 and 8.

Response: We thank the reviewer for the professional comments. We have revised the relevant terminology in the Manuscript. Thank you!

We revised the Manuscript into the following statements:

(Page 5) the average emission lifetime was 1.20 μ s with three lifetime components ($\tau_1 = 4.12 \mu$ s, $\tau_2 = 14.2 \mu$ s, and $\tau_3 = 0.21 \mu$ s; Figure S6)

(Page 6) The PL spectrum still displayed two lifetime components corresponding to Cu₈ and Cu₁₀ clusters (Figure S12B).

(Page 8) The Cu₈@Cu₁₀ crystal displayed a strong PL at 640 nm (QY = ~43.3 %; Figures 3E and S14) with a microsecond emission lifetime of 6.54 μ s with two lifetime components ($\tau_1=1.72 \mu$ s and $\tau_2=7.37 \mu$ s; Figures 3F and S15).

Finally, we would like to re-express our gratitude to this reviewer for his (her) professional feedback which has helped us to significantly improve this paper.

REVIEWERS' COMMENTS

Reviewer #3 (Remarks to the Author):

The authors have answered the questions satisfactorily and the manuscript can now be recommended for publication.